


# The importance of ecosystem adaptation on hydrological model predictions in response to climate change

Laurène J. E. Bouaziz[1,2], Emma E. Aalbers[3,4], Albrecht H. Weerts[2,5], Mark Hegnauer[2],
Hendrik Buiteveld[6], Rita Lammersen[6], Jasper Stam[6], Eric Sprokkereef[6], Hubert H. G. Savenije[1], and
Markus Hrachowitz[1]

[1]Department of Water Management, Faculty of Civil Engineering and Geosciences, Delft University of Technology, P.O. Box 5048, NL-2600 GA Delft, The Netherlands
[2]Department Catchment and Urban Hydrology, Deltares, Boussinesqweg 1, 2629 HV Delft, The Netherlands
[3]Royal Netherlands Meteorological Institute (KNMI), P.O. Box 201, 3730 AE De Bilt, the Netherlands
[4]Institute for Environmental Studies (IVM), Vrije Universiteit, Amsterdam, 1081 HV, the Netherlands
[5]Hydrology and Quantitative Water Management Group, Wageningen University and Research, P.O. Box 47, 6700 AA Wageningen, The Netherlands
[6]Ministry of Infrastructure and Water Management, Zuiderwagenplein 2, 8224 AD Lelystad, The Netherlands

**Correspondence:** Laurène Bouaziz (laurene.bouaziz@deltares.nl)

**Abstract.** To predict future hydrological behavior in a changing world, often use is made of models calibrated on past observations, disregarding that hydrological systems, hence model parameters, will change as well. Yet, ecosystems likely adjust their root-zone storage capacity, which is the key parameter of any hydrological system, in response to climate change. In addition, other species might become dominant, both under natural and anthropogenic influence. In this study, we propose a top-down

approach, which directly uses projected climate data to estimate how vegetation adapts its root-zone storage capacity at the catchment scale in response to changes in magnitude and seasonality of hydro-climatic variables. Additionally, the Budyko characteristics of different dominant ecosystems in sub-catchments are used to simulate the hydrological behavior of potential future land-use change, in a space-for-time exchange. We hypothesize that changes in the predicted hydrological response as a result of 2K global warming are more pronounced when explicitly considering changes in the sub-surface system properties

induced by vegetation adaptation to changing environmental conditions. We test our hypothesis in the Meuse basin in four scenarios designed to predict the hydrological response to 2K global warming in comparison to current-day conditions using a process-based hydrological model with (a) a stationary system, i.e. no changes in the root-zone storage capacity of vegetation and historical land use, (b) an adapted root-zone storage capacity in response to a changing climate but with historical land use, and (c,d) an adapted root-zone storage capacity considering two hypothetical changes in land use from coniferous plan-

tations/agriculture towards broadleaved forest and vice-versa. We found that the larger root-zone storage capacities (+34 %) in response to a more pronounced seasonality with drier summers under 2K global warming strongly alter seasonal patterns of the hydrological response, with an overall increase in mean annual evaporation (+4 %), a decrease in recharge (-6 %) and a decrease in streamflow (-7 %), compared to predictions with a stationary system. By integrating a time-dynamic representation of changing vegetation properties in hydrological models, we make a potential step towards more reliable hydrological

predictions under change.





# 1   Introduction

Hydrological models are required to provide robust short-term hydrological forecasts and long-term predictions of the impact of natural and human-induced change on the hydrological response. Common practice is to predict the future using a hydrological model calibrated to the past (Vaze et al., 2010; Blöschl and Montanari, 2010; Peel and Blöschl, 2011; Coron, 2013; Seibert and van Meerveld, 2016). For the near future, it seems acceptable to assume no fundamental change in the hydrological system, although we know that ecosystems, the manager of the hydrological system, have the capacity to adapt to climatic change (Savenije and Hrachowitz, 2017). For longer term predictions, it is therefore not correct to assume an unchanged system within a changing world. This raises the question on the robustness of hydrological predictions, especially in the context of climate change (Coron et al., 2012; Stephens et al., 2019).

For example, Merz et al. (2011) clearly shows the non-stationarity of hydrological model parameters when calibrating 273 Austrian catchments in subsequent 5-years periods between 1976 and 2006. Being the core parameter of any hydrological system, Merz et al. (2011) report almost a doubling of the root-zone storage capacity and this gradual increase is assumed to be related to changing climatic conditions, such as increased evaporation and drier conditions in the more recent years. The temporal variability of model parameters could also be attributed to uncertainties in input and model structure or inadequate calibration strategies. However, the observed trends in model parameters are also likely to reflect transient catchment conditions over the historical period.

Under continued global warming, precipitation and temperature extremes are expected to further increase and the hydrological cycle is likely to further accelerate (Allen et al., 2010; Kovats et al., 2014; Stephens et al., 2021). In addition, natural land cover change and anthropogenic activities of land-cover change and land-use management can substantially alter a catchment's water balance (Brown et al., 2005; Wagener, 2007; Fenicia et al., 2009; Jaramillo and Destouni, 2014; Nijzink et al., 2016a; Hrachowitz et al., 2020; Levia et al., 2020; Stephens et al., 2020, 2021). Considering the unprecedented speed of change, Milly et al. (2008) declared that stationarity is dead and no longer should serve as a default assumption in water management. He advocates the development of methods that quantify the non-stationarity of relevant hydrological variables.

However, understanding and representing non-stationarity is challenging due to the complex interactions and associated feedback between climate, vegetation, soils, ecosystems and humans (Seibert and van Meerveld, 2016; Stephens et al., 2020). The main methods to understand how changes in hydrological functioning relate to changes in catchment characteristics rely on paired watershed studies and hydrological modeling (Andréassian et al., 2003). In many modeling studies, a selection of one or more parameters are changed using values from literature in combination with adapted land-cover maps to (partly) reflect the characteristics of the altered system (Mao and Cherkauer, 2009; Buytaert and Beven, 2009; Pomeroy et al., 2012; Gao et al., 2015). Alternatively, Duethmann et al. (2020) uses satellite observations of vegetation indices to improve the representation of the surface resistance dynamics to calculate reference evaporation used in conceptual hydrological models over a historical record. A similar approach is applied by Fenicia et al. (2009) to account for changes in evaporation as a result of land-use management changes in the Meuse basin.



While these approaches are valuable to test the sensitivity of change on the hydrological response (Seibert and van Meerveld, 2016), they require an understanding of how catchment characteristics relate to model parameters. Yet, there is considerable uncertainty in a priori parameter estimation and the use of regionalization approaches (Wagener, 2007). Besides, the required data (e.g. future land-use maps or vegetation indices) may not be available in the context of climate change impact assessment (Duethmann et al., 2020). Instead, a way forward may be to develop robust top-down modeling approaches based on optimality principles by considering the co-evolution of soils, vegetation and climate in a holistic way (Blöschl and Montanari, 2010).

As complex and heterogeneous as landscapes may be across a diversity of climates, the long-term hydrological partitioning of a catchment is governed by a surprisingly simple and predictable relation, which relies on the available water and energy for evaporation (Turc, 1954; Mezentsev, 1955; Budyko, 1961; Fu, 1981; Zhang et al., 2004). The Budyko hypothesis, as often referred to, describes that mean annual evaporation over precipitation ($E_\mathrm{A}/P$) is mainly controlled by the aridity index, defined as the ratio of mean annual potential evaporation over precipitation ($E_\mathrm{P}/P$). However, Troch et al. (2013) found catchments to deviate from the Budyko hypothesis when exchanging climates across different catchments in a modeling experiment. Their results suggest that long-term hydrological partitioning results from the co-evolution of catchment properties and climate characteristics, including not only the aridity index but also climate seasonality, topography, vegetation and soils.

The combination of these other factors influencing the water balance partitioning besides the aridity index are explicitly considered in the $\omega$ parameter of the parametric description of the Budyko hypothesis (Fu, 1981; Zhang et al., 2004). Deviations from the Budyko curve suggest that different vegetation develops in different climates, along a different $\omega$ curve. If climate changes, catchments are likely not only to shift horizontally in the Budyko space as a result of a changing aridity index, but also vertically as a result of a changing vegetation cover (Jaramillo and Destouni, 2014). Vertical shifts within the Budyko space can also be related to vegetation-$CO_2$ interactions, e.g. $CO_2$ fertilization and improved water-use efficiency as a result of increasing $CO_2$ levels (Keenan et al., 2013; van der Velde et al., 2014; van Der Sleen et al., 2015; Ukkola et al., 2016; Jaramillo et al., 2018; Stephens et al., 2020).

The interdependence of climate seasonality, aridity index and vegetation to match the expectation from the Budyko curve was also demonstrated by Gentine et al. (2012); Donohue et al. (2012). Vegetation tends to efficiently adapt its root-zone storage capacity to satisfy canopy water demand. This implies that vegetation creates a larger buffer to survive dry spells when seasonal water supply and demand are out of phase, than in a climate where demand and supply are in phase. (Milly, 1994; Schymanski et al., 2008; Gerrits et al., 2009; Gentine et al., 2012; Gao et al., 2014). The root-zone storage capacity is, therefore, the key element regulating the partitioning of water fluxes in many terrestrial hydrological systems. In addition, not only natural changes to the environment, but also human interference with vegetation affect transpiration water demand and hence the root-zone storage capacity (Nijzink et al., 2016a; Hrachowitz et al., 2020).

Detailed observations of rooting-systems are very scarce in time and space and difficult to integrate to the catchment scale due to heterogeneity of landscapes (de Boer-Euser et al., 2016; Hrachowitz et al., 2020). Instead, the catchment-scale root-zone storage capacity is often estimated through calibration of a hydrological model. Other methods rely on optimality principles that maximize net primary production, carbon gain or transpiration (van Wijk and Bouten, 2001; Kleidon, 2004; Collins and Bras, 2007; Guswa, 2008; Speich et al., 2018). Alternatively, there is increasing evidence that the catchment-scale root-zone



storage capacity can be robustly and directly estimated from annual water deficits using water-balance data (Gao et al., 2014;
de Boer-Euser et al., 2016; Wang-Erlandsson et al., 2016; Nijzink et al., 2016a; Bouaziz et al., 2020; Hrachowitz et al., 2020).
However, it remains unclear how vegetation may adapt its root-zone storage capacity to climate change and how these changes
affect future hydrological behavior.

The objective of this study in the Meuse basin (Western Europe) is to quantify the sensitivity of the hydrological response to
potential changes in the root-zone storage capacity of vegetation in combination with land-use changes as a result of 2K global
warming. Using the Budyko framework, we first estimate changes in the long-term hydrological partitioning. To evaluate the
effect of land-use change under future conditions, we exchange space-for-time by connecting the spatially variable $\omega$ parameter
of the Budyko curve to different land uses. We then use water-balance data to estimate how the root-zone storage capacity may
adapt to increasing seasonal water deficits under climate change.

We hypothesize that changes in the predicted hydrological response as a result of 2K global warming in comparison to
current-day conditions are more pronounced when explicitly considering an adapted root-zone storage capacity to reflect
changes in the magnitude and seasonality of hydro-climatic variables as well as potential land-use changes. We test our hy-
pothesis using a process-based hydrological model and compare the difference in hydrological response when assuming (a) a
stationary system without changes in the root-zone storage capacity and historical land use, with three non-stationary systems
involving (b) an adapted root-zone storage capacity in response to climate change but no changes in land use, and (c,d) an
adapted root-zone storage capacity and two hypothetical land-use change scenarios.

## 2   Study area

### 2.1   Climate and landscape

The Meuse basin upstream of Borgharen, at the border between Belgium and the Netherlands, covers an area of approximately
21300 km$^2$ with an elevation ranging between 50 and 700 m and can be divided into three main zones (Fig. 1). The French
Southern part of the basin is characterized by relatively thick soils, broad valleys bottoms and gentle slopes underlain by
sedimentary consolidated rocks. Metamorphic rocks and relatively thin impermeable soils dominate the steeper Ardennes
Massif in Belgium. On the West bank of the Meuse in Wallonia, the lithology is characterized by porous chalk layers with deep
groundwater systems.

The Meuse is a rain-fed river with relatively short response times. Streamflow has a strong seasonality with low summer and
high winter streamflow reflecting the seasonality of potential evaporation, while precipitation is relatively uniformly distributed
throughout the year. The large storage capacity due to relatively thick soils in the French part of the basin increases the
hydrological memory of the system, implying a strong influence of winter precipitation on streamflow deficits in the subsequent
summer (de Wit et al., 2007). Snow is not a major component of the water balance, but snow melt can have a significant
influence during some events (de Boer-Euser, 2017; Bouaziz et al., 2021). Mean annual precipitation, potential evaporation
and streamflow is approximately 950 mm yr$^{-1}$, 580 mm yr$^{-1}$ and 407 mm yr$^{-1}$.





## 2.2 Land use

Land use in the basin consists of 35 % forest, 32 % agriculture, 21 % pasture and 9 % urban areas (Fig. 1, European Environment Agency, 2018). The large majority of forests in the French part of the basin is characterized as "old growth", here defined as forested area which has been continuously wooded since at least the middle of the 19th century (Cateau et al.,
2015). These broadleaved forests consist primarily of European oaks, sessile oak and beech (Institut National de l'Information Géographique et Forestière, 2019). In contrast, only 44 % of the 18th century Walloon forests of Belgium has remained from the original broadleaved forest, the rest being cleared for agriculture on high fertility soils in the North West (30 %) or converted to coniferous plantations (Scots pine, Norway spruce and Douglas-fir) on the poor soils of the Ardennes (26 %, Kervyn et al., 2018). The status of "old growth" forest does not exclude human disturbances, but assumes a relatively limited impact.
Soils are less disturbed and their structure and biochemical composition have been preserved for several centuries. This favors a high degree of biodiversity, which is a key element for the resilience of forest ecosystems to perturbations. In contrast, recent short-rotation plantations lack many of these characteristics. Particularly thick canopy plantations, such as the spruce and Douglas-fir, significantly alter the typical biodiversity of forests. Additionally, relatively higher evaporation water use is expected in these recent, short-rotation exotic plantations in comparison to older, more natural forests (Fenicia et al., 2009;
Stephens et al., 2021).

## 3   Data

### 3.1   Observed historical E-OBS climate data

The E-OBS dataset (v20.0e) includes daily precipitation, temperature and radiation fields for the period 1980-2018 at a 25 km$^2$ resolution (Cornes et al., 2018). The data are based on station data collated by the European Climate Assessment  Dataset (ECA&D) initiative. Temperature is downscaled using the digital elevation model and a fixed lapse rate of 0.0065°C m$^{-1}$.
Potential evaporation is estimated using the Makkink formula (Hooghart and Lablans, 1988). There is a relatively large underestimation of precipitation (> 20 %) in the E-OBS dataset in the center of the basin when compared to an operational dataset, which is based on local precipitation data provided by the Service Public de Wallonie for the period 2005-2017 (Bouaziz et al., 2020). A monthly bias-correction factor is applied to improve the consistency between both datasets (Sect. S1 of the
Supplement).

### 3.2   Simulated historical and 2K climate data

To study the impact of 2K global warming on the hydrological response of the Meuse basin, we use climate simulations of the historical period 1979-2018 and a 2K global warming simulation, provided by the Royal Netherlands Meteorological Institute (KNMI). The simulations are generated with the regional climate model KNMI-RACMO2 (van Meijgaard et al., 2008) at 12
km x 12 km resolution. RACMO2 uses the land surface scheme HTESSEL (Balsamo et al., 2009), which employs four soil





layers with a total depth of 2.9 m. Each land-grid cell includes separate tiles for high and low vegetation (16 vegetation types), bare soil, snow and intercepted water, for which the energy and water balances are solved individually.

The historical simulation uses ERA5 reanalysis data (Hersbach et al., 2020) as initial- and lateral boundary conditions. The 2K simulation is a so-called pseudo-global warming (PGW) simulation (e.g. Schär et al., 1996; Attema et al., 2014; Prein et al., 2017; Brogli et al., 2019), which is an alternative method to generate high-resolution climate change information. Instead of downscaling global climate model (GCM) projections, the historical period is re-simulated, but set against a warmer climate background by adding perturbations to the ERA5 initial- and boundary conditions. The perturbations represent the change in the mean climate state in a globally 2K warmer world, derived from a large initial condition GCM ensemble (Aalbers et al., 2018). The method minimizes biases in the mean climate state of the historical simulation, guaranties a realistic atmospheric circulation under both historical and 'future' conditions and increases the signal-to-noise ratio of the climate change response. A full description of the dataset is provided in Aalbers et al. (2021).

### 3.3 Streamflow

Streamflow data is available for 35 catchments nested within the Meuse basin upstream of Borgharen for the period 2005-2017 (Fig. 1, Service Public de Wallonie, 2018; Banque Hydro, 2018). The streamflow at Borgharen is a constructed time series which sums the observed streamflow of the Meuse at St Pieter and of the Albert Canal at Kanne to represent the total flow from the tributaries before part of it is extracted in the Albert Canal (de Wit et al., 2007).

### 4 Methods

To quantify the importance of reflecting ecosystem adaptation in hydrological models in response to climate change, the following stepwise approach is designed: (1) estimate the long-term runoff coefficient in a 2K warmer world from movements in the Budyko space as a result of a shift in aridity index and a potential shift in dominant land-use from broadleaved forests to coniferous plantation and agriculture and vice-versa by trading space-for-time in the Meuse basin; (2) estimate how the root-zone storage capacity adapts in response to a more pronounced seasonality with drier summers and changing dominant land use using the observed historical and estimated long-term runoff coefficient in a 2K warmer world with potential changes in land use; (3) calibrate a hydrological model with observed historical E-OBS climate data to represent current-day hydrological conditions; (4) test if the historical climate data simulated by the regional climate model leads to a plausible representation of current-day hydrological conditions; (5) run the hydrological model with the 2K climate data in four scenarios describing (a) a stationary system with historical root-zone storage capacity and historical land-use, (b) an adapted root-zone storage capacity in response to a changing climate but a historical land-use, (c,d) an adapted root-zone storage capacity and a shift in dominant land-use; and finally (6) compare the change in hydrological response between 2K and historical conditions for these four scenarios.





## 4.1 Changing climate, vegetation and land use

### 4.1.1 Long-term water balance framework for estimating the change in runoff coefficient

The long-term partitioning of precipitation ($P$) into evaporation ($E_A$) and streamflow ($Q$) is mainly controlled by the long-term aridity index (ratio of potential evaporation over precipitation, $E_P/P$), according to the Budyko hypothesis. To account for additional factors that influence the long-term hydrological partitioning, Fu (1981) introduces a parameter $\omega$ to encapsulate the combined influences of climate, soils, vegetation and topography (Equation 1).

$$\frac{E_A}{P} = 1 - \frac{Q}{P} = 1 + \frac{E_P}{P} - \left(1 + \left(\frac{E_P}{P}\right)^\omega\right)^{1/\omega} \tag{1}$$

We solve Equation 1 to determine the value of $\omega$ for each of the 35 catchments of the Meuse basin for observed historical conditions for the period 2005 to 2017 ($\omega_{\text{obs}}$), using the meteorological E-OBS data ($P_{\text{obs}}$ and $E_{\text{P,obs}}$) and observed streamflow ($Q_{\text{obs}}$). Assuming only a change in long-term mean climate conditions, i.e. aridity index, a catchment will move along its $\omega_{\text{obs}}$-parameterized curve from its original position ($p_1$) to a new position ($p_2$) due to the horizontal shift in aridity index ($\Delta E_P/\Delta P$, Fig. 2a). Here, we use the simulated historical and 2K climate data to determine how the change in potential evaporation ($\Delta E_P = E_{P,2K} - E_{P,\text{hist}}$) and precipitation ($\Delta P = P_{2K} - P_{\text{hist}}$) lead to a change in aridity index (Equation 2) and therefore in actual evaporation ($\Delta E_A = E_{A,2K} - E_{A,\text{hist}}$) and streamflow ($\Delta Q = Q_{2K} - Q_{\text{obs}}$), using Equation 1.

$$\left(\frac{E_P}{P}\right)_{2K} = \frac{E_{P,\text{obs}} + \Delta E_P}{P_{\text{obs}} + \Delta P} \tag{2}$$

However, land cover and vegetation are likely to also change in response to a changing climate, introducing an additional vertical shift ($\Delta\omega$) toward a position $p_3$ on a different $\omega_{\text{change}}$ curve (Fig. 2a). A downward vertical shift from $\omega_{\text{obs}}$ to $\omega_{\text{change}}$ indicates less water use for evaporation, as opposed to an upward shift for higher evaporative water use. These vertical shifts in $\omega$ values represent changes in drivers other than aridity index, including e.g. land cover, tree species, forest age, biomass growth and water use efficiency (Jaramillo et al., 2018).

To test the sensitivity of the hydrological response to a change in $\omega$ in addition to a change in aridity index, we consider two scenarios. The catchments with relatively high percentages of broadleaved forests (25-38% as in the French part of the basin) receive the $\omega$ values of catchments with relatively low percentages of broadleaved forests (1-12% as mainly in the Belgian Ardennes) and vice-versa (Fig. 1b). We denote $\omega_{\text{broadleaved}}$ for the catchments with relatively high percentages of broadleaved forests and $\omega_{\text{coniferous}}$ for the catchments where broadleaved forests were largely converted to coniferous plantations or agriculture. These scenarios are meant as a sensitivity analysis in the spirit of trading space-for-time (Singh et al., 2011) to evaluate the effect of potential future land-use management on the overall water balance.

When converting broadleaved forest to coniferous plantations, we expect an increase in water use for evaporation and therefore a vertical upward shift in $\omega$ values, as opposed to a downward shift when converting coniferous plantations to more natural broadleaved forests. The described vertical and horizontal movements in the Budyko space are used to estimate the projected





long-term runoff coefficients ($(Q/P)_{2K}$, Equation 3) as a result of, both, climate change but no changes in vegetation cover ($\omega_{\mathrm{obs}}$), and climate change in combination with changes in vegetation cover (by swapping $\omega_{\mathrm{broadleaved}}$ values to $\omega_{\mathrm{coniferous}}$ for a selection of catchments and vice-versa). The projected runoff coefficients are subsequently used to estimate changes in the root-zone storage capacity parameter (Sect. 4.1.2).

$$\left(\frac{Q}{P}\right)_{2\mathrm{K},\omega} = \frac{Q_{\mathrm{obs}}+\Delta Q}{P_{\mathrm{obs}}+\Delta P} = -\left(\left(\frac{E_{\mathrm{P}}}{P}\right)_{2\mathrm{K}} - \left(1+\left(\frac{E_{\mathrm{P}}}{P}\right)_{2\mathrm{K}}^{\omega}\right)^{1/\omega}\right) \tag{3}$$

### 4.1.2 Seasonal water balance for estimating the change in root-zone storage capacity $S_{\mathrm{R,max}}$

The root-zone storage capacity represents the maximum volume of water which can be held in pores of unsaturated soil
and which is accessible to roots of vegetation for transpiration. It is a key element controlling the hydrological response of hydrological systems. The long-term partitioning of precipitation into streamflow and evaporation in a changed climate can only match expectations as estimated from movements in the Budyko space (Sect. 4.1.1) if we consider that vegetation has adapted its root-zone storage capacity to offset hydro-climatic seasonality, by creating a buffer large enough to overcome dry spells (Gentine et al., 2012; Donohue et al., 2012; Gao et al., 2014). This is the main assumption underlying the water-balance
method to estimate the root-zone storage capacity at the catchment scale (Gao et al., 2014; Nijzink et al., 2016a; de Boer-Euser et al., 2016; Wang-Erlandsson et al., 2016; Bouaziz et al., 2020; Hrachowitz et al., 2020).

The water-balance method requires daily time series of precipitation, potential evaporation and a long-term runoff coefficient to estimate transpiration, as it depletes the root-zone storage during dry spells. Annual water deficits ($S_{\mathrm{R,def}}$) stored in the root-zone of vegetation to fulfill canopy water demand for transpiration are estimated on a daily time step as the cumulative sum of
daily effective precipitation ($P_{\mathrm{E}}$) minus transpiration ($E_{\mathrm{R}}$).

First, effective precipitation, i.e. the amount of precipitation that reaches the soil after interception evaporation ($E_{\mathrm{I}}$), is estimated by solving the water balance of a canopy storage ($S_{\mathrm{I}}$) with maximum interception storage capacity ($I_{\mathrm{max}}$, here taken as 2.0 mm), according to Equation 4.

$$P_{\mathrm{E}}(t) = P(t) - E_{\mathrm{I}}(t) - \frac{\mathrm{d}S_{\mathrm{I}}(t)}{\mathrm{d}t} \tag{4}$$

Next, the long-term transpiration $\overline{E}_{\mathrm{R}}$ is estimated from the long-term water balance, using mean annual streamflow and effective precipitation ($\overline{Q}$ and $\overline{P}_{\mathrm{E}}$, all in mm yr$^{-1}$, Equation 5), assuming negligible changes in storage and intercatchment groundwater flows (catchments where $\overline{E}_{\mathrm{A}} = \overline{P} - \overline{Q} < \overline{E}_{\mathrm{P}}$).

$$\overline{E}_{\mathrm{R}} \approx \overline{P}_{\mathrm{E}} - \overline{Q} \tag{5}$$





The long-term transpiration $\overline{E}_\mathrm{R}$ is subsequently scaled to daily transpiration estimates $E_\mathrm{R}$, using the daily signal of potential

evaporation minus interception evaporation, according to Equation 6 (Nijzink et al., 2016a; Bouaziz et al., 2020).

$$E_\mathrm{R}(t) = (E_\mathrm{P}(t) - E_\mathrm{I}(t)) \cdot \frac{\overline{E}_\mathrm{R}}{(\overline{E}_\mathrm{P} - \overline{E}_\mathrm{I})} \tag{6}$$

The maximum annual storage deficits can then be derived from the cumulative difference of effective precipitation ($P_\mathrm{E}$) and

transpiration ($E_\mathrm{R}$), assuming an "infinite" storage, according to Equation 7 and illustrated in Fig. 2b. For each year, $S_\mathrm{R,def}$

represents the amount of water accessible to the roots of vegetation for transpiration during a dry period. Storage deficits are

assumed to be zero at the end of the wet period ($T_0$, here April) and increase when transpiration exceeds effective precipitation

during dry periods, until they become zero again ($T_1$) when excess precipitation is assumed to drain away as direct runoff or

recharge.

$$S_\mathrm{R,def}(t) = \min \int_{T_0}^{T_1} (P_\mathrm{E}(t) - E_\mathrm{R}(t)) \mathrm{d}t \tag{7}$$

By fitting the annual maximum storage deficits to the extreme value distribution of Gumbel, the root-zone storage capacity

at the catchment scale $S_\mathrm{R,max}$ can be derived for various return periods. Previous studies used a return period of 20 years

for forested areas, meaning that forests develop root systems to survive droughts with a return period of $\sim$20 years (Nijzink

et al., 2016a; de Boer-Euser et al., 2016; Hrachowitz et al., 2020). The root-zone storage capacity of cropland and grasslands

is assumed to correspond to deficits with lower return periods of $\sim$2 years (Wang-Erlandsson et al., 2016). It should be noted

that the methodology assumes that vegetation taps its water from the unsaturated zone and not from the groundwater.


Using the above described methodology, we determine several sets of $S_\mathrm{R,max}$ values for each of the 35 catchments of

the Meuse basin to represent the historical and adapted root-zone storage capacity in response to a changing climate and

changing/historical land-use, using historical climate observations (E-OBS) and the historical and 2K climate simulations

(Table 1).

**$S_\mathrm{R,max,A}$: Historical root-zone storage capacity from historical land use and observed historical climate**

The first set is $S_\mathrm{R,max,A}$, which represents the historical meteorological and land-use conditions, derived from observed his-

torical E-OBS data ($P_\mathrm{obs}$, $E_\mathrm{P,obs}$ for the period 1980-2018) and observed streamflow data ($Q_\mathrm{obs}$ for the period 2005-2017).

$S_\mathrm{R,max,A}$ is used as parameter for three model runs, each forced with a different dataset: historical E-OBS observations, simu-

lated historical and 2K climate data (Sect. 4.4).

In this study, we assume that the observed E-OBS historical climate data is the best available estimate of current-day climate

conditions and use this data to estimate historical root-zone storage capacities $S_\mathrm{R,max,A}$ and to calibrate the hydrological model

(Sect. 4.3). The simulated historical climate data is also required to enable a fair comparison with the simulated 2K climate





data, as they are both generated with the regional climate model. Despite potential biases in the climate model simulations compared to the observed historical data (here, E-OBS), we do not apply a formal bias-correction of the climate data which may alter the relations between variables in climate models (Ehret et al., 2012). Instead, we force the hydrological model with the native simulated historical climate data, but use the previously determined $S_{\mathrm{R,max,A}}$ parameter. An alternative approach would have been to estimate the root-zone storage capacities using the simulated historical climate data, to directly correct for potential biases in the climate data in the estimation of the root-zone storage capacity parameter but with the downside of affecting spatial patterns across catchments. For comparison, this analysis is performed in Sect. S2 of the Supplement.

### $S_{\mathrm{R,max,B}}$: Adapted root-zone storage capacity from historical land use and 2K climate

We then estimate the root-zone storage capacity $S_{\mathrm{R,max,B}}$ based on the 2K climate and historical land use to reflect vegetation adaptation to changing climatic conditions such as differences in seasonality, aridity index (Equation 2) and the resulting runoff coefficient (Equation 3), but under the assumption that the vegetation cover remains unchanged. To account for differences in the observed and simulated historical climate data, $S_{\mathrm{R,max,B}}$ is determined by imposing the difference in storage deficits derived from the 2K and historical climate simulations ($S_{\mathrm{R,def,2K}} - S_{\mathrm{R,def,hist}}$) on the observed storage deficit derived with E-OBS data $S_{\mathrm{R,def,obs}}$, as shown in Table 1.

### $S_{\mathrm{R,max,C}}$: Adapted root-zone storage capacity from land use conversion to broadleaved forest and 2K climate

Subsequently, the root-zone storage capacity is estimated for the 2K climate under two land-use change scenarios, considering that if climate changes, a different vegetation cover might become dominant under natural and anthropogenic influence (Table 1). Making use of a space-for-time exchange, we connect the spatially variable $\omega$ parameter of the Budyko curve to different land-use categories and use these to evaluate future land-use scenarios. Here, the catchments are categorized according to the areal fraction of broadleaved forest in a catchment (Fig. 1b). In the first scenario, land use in the catchments with mainly coniferous plantations and agriculture (as mainly in the Belgian Ardennes, Sect. 2.2 and Fig. 1) is assumed to be converted to broadleaved forest, using sampled $\omega_{\mathrm{broadleaved}}$ values of catchments within the French part of the basin to estimate the 2K runoff coefficient with Equation 3. The sampling is performed because the variability in $\omega$ values in each category is also influenced by other factors besides the dominant presence of broadleaved forest. The resulting $S_{\mathrm{R,max,C}}$ thus represents an adapted root-zone storage capacity in response to climate change and land-use conversion to broadleaved forest.

### $S_{\mathrm{R,max,D}}$: Adapted root-zone storage capacity from land use conversion to coniferous plantation and agriculture and 2K climate

Similarly, the adapted root-zone storage capacity $S_{\mathrm{R,max,D}}$ is estimated for the 2K climate in a land-use scenario where the broadleaved forest in the French part of the basin are converted to coniferous plantations and agriculture, using sampled $\omega_{\mathrm{coniferous}}$ values of catchments in the Belgian Ardennes.



## 4.2 wflow_FLEX-Topo hydrological model

The wflow_FLEX-Topo model (de Boer-Euser, 2017; Schellekens et al., 2020) is a fully distributed process-based model,
which uses different model structures for a selection of Hydrological Response Units (HRUs) to represent the spatial variability
of hydrological processes. Here, we develop a model with three HRUs for wetlands, hillslopes and plateaus connected through
their groundwater storage (schematized in Fig. 3 and model equations in Sect. S3 of the Supplement; Savenije, 2010; de Boer-
Euser, 2017). Thresholds of 5.9 m for the Height Above the Nearest Drainage (HAND Rennó et al., 2008) and 0.129 for
slope are used to delineate the three HRUs (Gharari et al., 2011) using the MERIT hydro dataset at ∼60 m x 90 m resolution
(Yamazaki et al., 2019). Given the high proportion of forest on hillslope and of agriculture on plateau, we here associated
hillslope with forest and agriculture with plateau, using the CORINE land cover data (European Environment Agency, 2018).
The areal fraction of each HRU are then derived for each cell at the model resolution of 0.00833° (or ∼600 m x 900 m). The
model includes snow, interception, root-zone, fast and slow storage components. Streamflow is routed through the upscaled
river network at the model resolution (Eilander et al., 2020) with the kinematic wave approach. Similar implementations of that
model were previously successfully used in a wide variety of environments (e.g. Gharari et al., 2013; Gao et al., 2014; Nijzink
et al., 2016b; de Boer-Euser, 2017; Hulsman et al., 2020; Bouaziz et al., 2021).

## 4.3 Model calibration and evaluation

### 4.3.1 Calibration and evaluation using the observed historical E-OBS climate data

The wflow_FLEX-Topo model is calibrated using streamflow at Borgharen and the observed historical E-OBS meteorological
forcing data for the period 2007-2011, using 2005-2006 as warm-up years. The observed historical E-OBS dataset is used for
calibration of the model as it is assumed to most closely represent current-day conditions. The parameter space is explored with
a Monte Carlo strategy, sampling 10000 realizations from uniform prior parameter distributions (Sect. S4 of the Supplement).
The limited number of samples is due to the high computational resources required to run the distributed model. However, our
aim is not to find the "optimal" parameter set, but rather to retain an ensemble of plausible parameter sets based on a multi-
objective calibration strategy (Hulsman et al., 2019). To best reflect different aspects of the hydrograph, including high flows,
low flows and medium-term partitioning of precipitation into drainage and evaporation, parameter sets are selected based on
their ability to simultaneously and adequately represent four objective functions, including the Nash-Sutcliffe efficiencies of
streamflow, the logarithm of streamflow and, monthly runoff coefficients as well as the Kling-Gupta efficiency of streamflow.
Only parameter sets that exceed a performance threshold of 0.9 for each metric are retained as feasible. The root-zone storage
capacity parameter $S_{R,max,A}$ is a fixed parameter, which is derived from annual maximum storage deficits with a return period
of 2 years for the wetland and plateaus HRUs and 20 years for the hillslopes HRU (Sect. 4.1.2). Next, model performance
is evaluated in the 2012-2017 post-calibration period using the same performance metrics, a visual inspection of the hydro-
graphs and the mean monthly streamflow regime. The performance metrics are also evaluated for the 34 remaining nested
sub-catchments.





### 4.3.2 Evaluation using the simulated historical climate data

The performance of the calibrated model for the ensemble of retained parameter sets is also evaluated when the model is forced with the simulated historical climate data, using $S_{\mathrm{R,max,A}}$ for the root-zone storage capacity parameter. This is the reference historical run against which the relative effect of 2K global warming is evaluated for different scenarios (Fig. 4 and Sect. 4.4). In addition, in Sect. S2 of the Supplement, we evaluate the performance of the calibrated model forced with the simulated historical climate data but with a root-zone storage capacity parameter derived directly from this data. While this alternative approach enables to correct for potential biases in the simulated historical climate data directly in the estimation of the root-zone storage capacity parameter, it may also affect the spatial patterns of this parameter across catchments.

### 4.4 Hydrological change evaluation

We then force the calibrated wflow_FLEX-Topo model for the ensemble of retained parameter sets with the 2K climate data in four scenarios each using a different root-zone storage capacity parameter to represent either stationary or adapted conditions in response to a changing climate and land use (Fig. 4 and Sect. 4.1.2). The difference between the modelled historical hydrological response (1980-2018) and the hydrological responses predicted by each of the four model scenarios based on the 2K climate is evaluated in terms of runoff coefficient, evaporative index, annual statistics (runoff coefficient, evaporative index, mean, maximum, minimum 7-days streamflow and median volume deficit below the 90[th] percentile reference streamflow), and monthly patterns of flux and state variables (streamflow, evaporation, root-zone storage, groundwater storage) for a hypothetical 38-year period.

**Scenario 2K$_{\mathrm{A}}$: Historical land use and historical root-zone storage capacity ($S_{\mathrm{R,max,A}}$)**

In scenario 2K$_{\mathrm{A}}$ (Fig. 4), we assume an unchanged land use and that vegetation has not adapted its root-zone storage capacity to the aridity and seasonality of the 2K climate. This scenario implies stationarity of model parameters by using $S_{\mathrm{R,max,A}}$ in both the historical and 2K runs, a common assumption of many climate change impact assessment studies (Booij, 2005; de Wit et al., 2007; Prudhomme et al., 2014; Hakala et al., 2019; Brunner et al., 2019; Gao et al., 2020; Rottler et al., 2020). This is the benchmark scenario against which we compare the hydrological response considering non-stationarity of the system, as in the following three scenarios.

**Scenario 2K$_{\mathrm{B}}$: Historical land use and 2K climate adapted root-zone storage capacity ($S_{\mathrm{R,max,B}}$)**

In scenario 2K$_{\mathrm{B}}$ (Fig. 4), we again assume an unchanged land use ($\omega_{\mathrm{obs}}$). However, we assume that vegetation has adapted its root-zone storage capacity to the aridity and seasonality of the 2K climate conditions by selecting $S_{\mathrm{R,max,B}}$ as parameter for the 2K model run, while the historical $S_{\mathrm{R,max,A}}$ is used as parameter in the historical run.





**Scenario 2K$_\mathrm{C}$: Land-use conversion to broadleaved forest and 2K climate adapted root-zone storage capacity ($S_\mathrm{R,max,C}$)**

In scenario 2K$_\mathrm{C}$ (Fig. 4), we adapt the root-zone storage capacity to the changing aridity index and seasonality of the 2K climate. Additionally, we assume a change in vegetation cover for the catchments located mainly in the Belgian Ardennes and dominated by coniferous plantation and agriculture to a land use of broadleaved forest as in the French part of the basin. For this purpose, $S_\mathrm{R,max,C}$ is used as parameter in the model run forced with the 2K climate, while $S_\mathrm{R,max,A}$ is used as parameter in the historical run.

**Scenario 2K$_\mathrm{D}$: Land-use conversion to coniferous plantation and agriculture and 2K climate adapted root-zone storage capacity ($S_\mathrm{R,max,D}$)**

In scenario 2K$_\mathrm{D}$ (Fig. 4), the approach of scenario 2K$_\mathrm{C}$ is repeated. However, now the broadleaved forest in the French catchments are assumed to be converted to coniferous plantations or agriculture as in the Belgian Ardennes. The parameter $S_\mathrm{R,max,D}$ is used in the model run forced with the 2K climate, while $S_\mathrm{R,max,A}$ is used in the historical run.

# 5   Results

## 5.1   Adapted root-zone storage capacity $S_\mathrm{R,max}$ from long-term and seasonal water balances and changing land use

### 5.1.1   Long-term water balance characteristics across catchments

In solving the parametric Budyko curve (Equation 1) for the 35 catchments of the Meuse basin using historical E-OBS data and observed streamflow (Fig. 5a), we found that $\omega_\mathrm{obs}$ values tend to be lower (median of $2.43 \pm 0.48$) for catchments with
relatively high percentages of broadleaved forests (25-38 % as in the French part of the basin) as compared to catchments with relatively low percentages of broadleaved forests (1-12 % as in the Belgian part of the catchment) with median $\omega$ values of $3.04 \pm 0.54$, as shown in Fig. 5b. Higher values of $\omega$ for a same aridity index indicate more water use for evaporation, which is likely related to the increased water use of relatively young coniferous plantations and agriculture as opposed to older broadleaved forests (Fenicia et al., 2009; Teuling et al., 2019).

### 5.1.2   $S_\mathrm{R,max,A}$ from historical land use and historical climate

The root-zone storage capacity $S_\mathrm{R,max,A}$ derived with observed historical E-OBS climate data and observed streamflow is estimated at values of $101 \pm 17$ mm and $169 \pm 24$ mm across all study catchments for a 2 year and 20 year return period, respectively (Fig. 5c).

      If instead the simulated historical climate data is used to derive the root-zone storage capacity, this results in slightly higher
values with $110 \pm 18$ mm and $180 \pm 28$ mm for 2 and 20 year return periods, respectively. This overestimation of about +7 % is due to the higher precipitation (on average +9 %) in the simulated historical climate data compared to the observed E-OBS





historical data, which leads to relatively lower runoff coefficients and therefore larger evaporative indices and storage deficits in the water balance calculation of the root-zone storage capacity (Sect. S2 of the Supplement).

### 5.1.3 $S_{\mathrm{R,max,B}}$ from historical land use and 2K climate

The adapted root-zone storage capacity $S_{\mathrm{R,max,B}}$, in response to changing climate conditions and an unchanged land use, strongly increases with respect to historical conditions ($S_{\mathrm{R,max,A}}$) with estimated values of $129 \pm 18$ mm (+28 %) and $227 \pm 27$ mm (+34 %) for return periods of 2 year and 20 year, respectively (Fig. 5c). This strong increase is explained by larger storage deficits during summer due to an increase of about +10 % in summer potential evaporation in the 2K climate and, therefore, a more pronounced seasonality (Fig. 2b). In contrast, the change in aridity index between the historical and

2K climate simulations is relatively small with a median of +0.01 across all study catchments. This can be explained by a simultaneous increase in mean annual precipitation (+5 %) and potential evaporation (+7 %) on average over the basin area in the 2K climate compared to the simulated historical climate data. This increase in precipitation mostly occurs during the winter half year (Nov-Apr). In contrast, there is a relatively large variability in precipitation change in summer, characterized by years with wetter and drier summers.

### 400 5.1.4 $S_{\mathrm{R,max,C}}$ from 2K climate and adapted land use to broadleaved forest

The adapted root-zone storage capacity $S_{\mathrm{R,max,C}}$, in response to changing climate conditions and a land use conversion from coniferous plantation and agriculture to broadleaved forest, results in estimated values of $125 \pm 17$ mm and $219 \pm 27$ mm for return periods of 2 year and 20 year, respectively (Fig. 5c). These values are almost similar to $S_{\mathrm{R,max,B}}$, with a difference of about -3 %. This small decrease is in line with the expected reduced water use of broadleaved forests compared to coniferous

plantations.

### 5.1.5 $S_{\mathrm{R,max,D}}$ from 2K climate and adapted land use to coniferous plantations and agriculture

In contrast, the root-zone storage capacity $S_{\mathrm{R,max,D}}$, in response to changing climate conditions and a conversion of broadleaved forest to coniferous plantation, result in estimated values of $140 \pm 22$ mm and $243 \pm 35$ mm for return periods of 2 year and 20 year, respectively (Fig. 5c). This corresponds to an increase of +9 % and +7 % for both return periods in comparison with

$S_{\mathrm{R,max,B}}$, which does not consider additional land-use changes.

The difference in root-zone storage capacity between the 2K and historical climate simulations as a result of a changing climate (aridity and seasonality) is larger (+58 mm or +34 % for a return period of 20 years) than the difference between root-zone storage capacity for a changing climate and additional changes in land use (-8 mm or -3 % for $S_{\mathrm{R,max,C}}$ and +16 mm or +7 % for $S_{\mathrm{R,max,D}}$). This indicates that with the assumed land-use change in scenarios $2\mathrm{K_C}$ and $2\mathrm{K_D}$, the strong increase

in water demand during summer as a result of a more pronounced seasonality has greater impact on the estimation of the root-zone storage capacity than a change in $\omega$ values. However, note that land use is changed in only part of the catchment for





both land use change scenarios and that it is plausible to assume that more pronounced changes in land use will reinforce the observed effects.

## 5.2 Model evaluation (historical period)

### 5.2.1 Model forced with observed historical climate data

The ensemble of parameter sets retained as feasible after calibration mimics the observed hydrograph at Borgharen relatively well for the evaluation period (Fig. 6a). Also the seasonal streamflow regime is relatively well reproduced by the model, except for a slight underestimation of about -9 % in the first half year (Fig. 6b). The four objective functions show a relatively similar performance during calibration and evaluation with median values of approximately 0.93 and 0.78 at Borgharen and for the ensemble of nested catchments of the Meuse, respectively (Fig. 7a,b).

### 5.2.2 Model forced with simulated historical climate data

When the calibrated model is instead forced with the simulated historical climate data, peaks are slightly overestimated in comparison to the model run forced with the observed historical E-OBS data (Fig. 6c). This is due to the on average +9 % overestimation of precipitation in the simulated historical climate data compared to the observed historical E-OBS climate data. This precipitation overestimation results in an overestimation of about +12 % of modeled mean monthly streamflow during the wettest months (Fig. 6d). The streamflow model performance at Borgharen slightly decreases when the simulated historical climate data is used instead of E-OBS, but median values across the ensemble of feasible parameter sets are still above 0.77 for each of the objective functions (Fig. 7c). Although a decrease in model performance is found in a few nested catchments, the performance in the ensemble of nested catchments of the Meuse remains relatively high with median values of around 0.67 (Fig. 7d). The results of the model run forced with the simulated historical data and with the root-zone storage capacity parameter derived directly from this data show a relatively similar behavior, as further detailed in Sect. S2 of the Supplement.

The calibrated model forced with the simulated historical climate data shows a plausible behavior with respect to observed streamflow and is also close to the performance achieved with the observed historical E-OBS climate data. This is important because the effect of the 2K climate on the hydrological response is evaluated with respect to the model run forced with the simulated historical climate data, as they are both generated with the regional climate model. Therefore, the relatively high model performance in the evaluation period enable us to use the retained parameter sets from the calibration with E-OBS data for the subsequent analyses with the simulated historical and 2K climate data.





### 5.3 Hydrological change evaluation (2K warmer climate)

#### 5.3.1 Scenario $2K_A$: Stationarity with historical land use and historical root-zone storage capacity ($S_{R,max,A}$)

In the $2K_A$ scenario, representing a stationary system with identical parameters in the historical and 2K climate, runoff coefficients are projected to increase with a median of +3 %, the evaporative index ($E_A/P$) decreases with a median of -2 % and mean annual streamflow increases with a median of +7 %. Maximum annual streamflow is also projected to increase with a median of about +5 %, while the median change in annual minimum of 7-days mean streamflow remains close to zero. The median annual deficit volume below the 90[th] percentile historical streamflow increases with +10 %, as shown in Fig. 8.

Streamflow is projected to increase from December until August with +8 % and decrease between September and November with -7 %. In the months where evaporation demand exceeds precipitation, the root-zone soil moisture decreases, with a maximum of -22 % in September. Actual evaporation increases throughout the year with +3 % except in July and August (-4 %) when the availability of water in the root-zone of vegetation is not sufficient to supply canopy water demand. Recharge to the groundwater storage increases with approximately +5 % in all months except November, as shown in Fig. 9.

#### 5.3.2 Scenario $2K_B$: Historical land use and adapted root-zone storage capacity ($S_{R,max,B}$)

Changes are substantially different in the $2K_B$ scenario which considers that the root-zone storage capacity of vegetation has adapted to the change in aridity and seasonality of the 2K climate. Runoff coefficients are instead projected to decrease with a median of -2 %, while the evaporative index increases with a median of +2 % and the median change of mean annual streamflow is close to zero (Fig. 8). Also the median change of, both, annual maximum streamflow and minimum 7-days mean streamflow remain close to zero. However, there is a substantial increase of +38 % in median annual deficit volume below the 90[th] percentile historical streamflow. This result suggests that while the minimum streamflow remains relatively similar, the length of the low flow period strongly increases if we consider that the root-zone storage capacity has adapted to the 2K climate (Fig. 8).

Seasonal changes indicate a decrease in streamflow of -19% between September and November, which is longer and more pronounced than in the $2K_A$ scenario (Fig. 9). The root-zone soil moisture increases throughout the year with an average of +34 % due to the larger root-zone storage capacities. Actual evaporation is no longer reduced as a result of moisture stress in the root-zone and strongly increases with approximately +7 % from May to October to supply canopy water demand. The increase in evaporation during summer strongly reduces the groundwater recharge with -5 % from October to February (Fig. 9).

#### 5.3.3 Scenario $2K_C$: Land-use conversion to broadleaved forest and adapted root-zone storage capacity ($S_{R,max,C}$)

The predicted hydrological response in the $2K_C$ scenario is very similar to the response of the $2K_B$ scenario, despite considering additional changes in the root-zone storage capacity as a result of a land-use conversion from coniferous plantations and agriculture to broadleaved forest (Figs. 8 and 9). This is in line with the limited differences in root-zone storage capacities of approximately +3 % between both scenarios (Sect. 5.1).





### 5.3.4   Scenario $2K_D$: Land-use conversion to coniferous plantations and agriculture and adapted root-zone storage capacity ($S_{R,max,D}$)

In contrast, the change in hydrological response is most pronounced for the scenario $S_{R,max,D}$, which considers land use conversion of the broadleaved forests in the French part of the basin to coniferous plantations and agriculture (Figs. 8 and 9). Runoff coefficients decrease with a median of -4 %, while the evaporative index increases with a median value of +4 % and mean annual streamflow decreases with a median of -2 %. If the median change in streamflow extremes remains relatively close to zero, there is a strong increase of +54 % in the median annual deficit volume, suggesting a strong increase in the length of the low flow period (Fig. 8).

Streamflow decreases from August to January with an average of -23 % and evaporation strongly increases from May to October with an average of +9 %. This increased evaporation during summer further reduces recharge from October to February with -7 % (Fig. 9). In comparison with the hydrological response of scenario $2K_B$, the additional land-use conversion in scenario $2K_D$ results in relatively similar patterns of change but with an additional +2 % increase in evaporation, -2 % decrease in streamflow and -2 % decrease in recharge on average throughout the year.

### 5.3.5   Stationary versus adaptive ecosystems

There is a difference of -7 % in the change of mean annual streamflow between the scenarios $2K_B$, $2K_C$, $2K_D$ with adaptive ecosystems and the stationary $2K_A$ scenario. Additionally, the scenarios with adaptive ecosystems show a more pronounced decrease in streamflow from September to January and a delay in the occurrence of the lowest streamflow from September to October. The change in mean annual actual evaporation is approximately +4 % higher in the scenarios with adaptive ecosystems and the increase mainly occurs between May and October. Instead of a year-round increase in recharge in the $2K_A$ scenario, there is a decrease in winter recharge in the three other scenarios, resulting in a mean annual difference of -6 % between the scenarios with ecosystem adaption and the stationary scenario $2K_A$. Hence, the hydrological response in the 2K climate of the stationary scenario $2K_A$ is substantially different from the responses of the three scenarios $2K_B$, $2K_C$, $2K_D$, which consider a change in the root-zone storage capacity to reflect ecosystem adaptation in response to climate change.

## 6   Discussion

### 6.1   Implications

The hydrological response under 2K global warming with respect to historical conditions shows distinct patterns of change if we explicitly consider the non-stationarity of climate-vegetation interactions in a process-based hydrological model. We implement a dynamic root-zone storage capacity parameter, which is directly inferred from long-term and seasonal water balances of historical observations in combination with simulated historical and future climate conditions. A time-dynamic parameterization of the root-zone storage capacity was previously introduced by Nijzink et al. (2016a) in the context of deforestation, while it was implemented by Speich et al. (2020) in the context of climate change. In the latter study, forest growth in response to





climate change leads to a six times higher reduction of streamflow if a dynamic representation of, both, the Leaf Area Index and the root-zone storage capacity is implemented as opposed to a study in which only the Leaf Area Index varies (Schattan et al., 2013). These results are more pronounced than our findings but point towards the same direction of change. While Speich et al. (2020) combine a forest landscape model with a hydrological model to simultaneously represent the spatio-temporal forest and

water balance dynamics, we rely on a simpler approach of movements in the Budyko framework to include potential land-use change.

The concept of trading space-for-time, which uses space as a proxy for time (Singh et al., 2011) could be further explored by selecting a region outside the Meuse basin with a current climate similar to the projected climate. This approach is also commonly referred to as climate-analogue mapping, i.e. statistical techniques to quantify the similarity between the future

climate of a given location and the current climate of another location (Rohat et al., 2018; Bastin et al., 2019; Fitzpatrick and Dunn, 2019). Finding a climate analogue for future projections in present conditions, may allow us to estimate future $\omega$ or root-zone storage capacity values in a region where the future climate may resemble today's climate elsewhere. These methods are intuitive but not straightforward, as they rely on the selection and combination of relevant climate variables and their relation with vegetation, despite non-linear vegetation responses to climate change (Reu et al., 2014).

In comparing several scenarios for the root-zone storage capacity parameter, we include some form of system representation uncertainty, which improves our understanding in the modeled changes by placing them in a broader context (Blöschl and Montanari, 2010). Actual evaporation in the study catchments is projected to decrease if the historical root-zone storage capacity is used as a result of moisture stress in the root-zone of vegetation. In contrast, the increased water demand during summer is met when we assume that vegetation has adapted its root-zone storage capacity. This is an indication of disagreements among

model representations on processes that become relevant in the future, in line with findings of Magand et al. (2015); Melsen and Guse (2020).

The impact of climate change on low flows in the Meuse basin has been previously studied by de Wit et al. (2007). Using simulations from regional climate models which project wetter winters and drier summers, they question if the increase in winter precipitation reduces the occurrence of summer low flows due to an increase in groundwater recharge. However, they

were unable to address this question with their model due to its poor low-flow performance. Our results indicate an increase in groundwater recharge during winter if the historical root-zone storage capacity parameter is used, as opposed to a decrease for the models with a time dynamic root-zone storage capacity, as a result of an increased water demand for evaporation during summer. This further emphasizes the major impact of vegetation in regulating the water cycle (Luo et al., 2020; Wang et al., 2020; Stephens et al., 2021).

The land surface scheme HTESSEL, that is used in the regional climate model RACMO2 to generate the historical and 2K climate simulations, assumes, as most land surface models, a fixed root-zone storage capacity. Ideally, this discrepancy between the land surface model and the hydrological model could be reduced by updating the adapted root-zone storage capacity from one model to the other in several iteration steps, thereby including soil moisture - atmosphere feedback mechanisms.





## 6.2 Limitations and knowledge gaps

Our study relies on the assumption that vegetation has had the time to adapt its root-zone storage capacity in a changing climate. Yet, considering the unprecedented scale and rate of change (Gleeson et al., 2020), it is unclear how ecosystems will cope with climate change, also considering the impact of storms, heatwaves, fires and biotic infestations as a result of water stress on forest ecosystems (Lebourgeois and Mérian, 2011; Allen et al., 2010; Latte et al., 2017; Stephens et al., 2021). Additionally, when exposed to different environmental conditions, ecosystems may adapt their behavior by reducing or increasing their water use to the water availability (Zhang et al., 2020). Similarly, direct human interventions, such as the conversion of natural forests to fast-growing monoculture plantations in many parts of the world has significantly altered forests, making them more susceptible and vulnerable to disturbances (Schelhaas et al., 2003; Levia et al., 2020). However, humans also have the ability to positively influence the water cycle through vegetation, by promoting sustainable agricultural practices and integrated forest management with a simultaneous focus on biodiversity, recreation and timber production. Additionally, the conversion of exotic to native species may also increase biodiversity and the resilience of ecosystems to climate change (Klingen, 2017). However, these processes are slow, implying that current management practices shape the forests of decades and centuries to come in an uncertain future. Increasing our understanding on how to include these changes in hydrological models to reliably quantify their impact is a way forward in the development of strategies to mitigate the adverse effects of climate change.

We quantify the changes in the hydrological response as a result of a changing climate in combination with several land use scenarios (historical, conversion of broadleaved forests to coniferous plantations and agriculture and vice-versa). The relatively limited additional effects of land-use change on the hydrological response should be understood in the context of the relatively limited areal fraction under potential land-use conversion. The land-use changes are integrated in the root-zone storage capacity as single parameter. However, climate and land use changes likely affect other aspects of catchment functioning (Seibert and van Meerveld, 2016). For example, changes in the maximum interception storage capacity (Calder et al., 2003) are not explicitly considered in the estimation of the adapted root-zone storage capacities, as the impact was shown to be relatively minor in Bouaziz et al. (2020). Additional effects of soil compaction and artificial drainage on peak flows as a result of future land conversion (Buytaert and Beven, 2009; Seibert and van Meerveld, 2016) are difficult to quantify but may partly be captured in the changed $\omega$ values.

In a first step towards temporally adaptive models, and trading space-for-time for different land-use scenarios, we did not consider any additional vertical movements in the Budyko space due to the effects of increasing $CO_2$ levels in terms of increased productivity through fertilization, on the one hand, or water use efficiency on the other hand (Keenan et al., 2013; van der Velde et al., 2014; van Der Sleen et al., 2015; Ukkola et al., 2016; Jaramillo et al., 2018; Yang et al., 2019; Stephens et al., 2020), as these effects remain problematic to quantify in a meaningful way. Neither did we, for the same reason, consider how the relatively high $\omega$ values may be related to intercatchment groundwater losses (Bouaziz et al., 2018). Note that as our analyses should be understood in the context of a sensitivity analyses of the impact of potential additional vertical shifts in the Budyko space as a result of a changing land use (Fig. 2), the potential effects on groundwater losses on the results are likely to be minor.



In addition, we performed a limited calibration of the hydrological model to retain an ensemble of plausible solutions and only used a single climate simulation despite the uncertainty in initial and boundary conditions of regional climate models. Our

analyses are intended as a proof-of-concept to introduce a top-down methodology to quantify non-stationarity of the root-zone storage capacity parameter through optimal use of projected climate data, rather than a comprehensive climate change impact assessment of the Meuse basin.

## 7 Conclusions

Understanding non-stationarity of hydrological systems under climate and environmental changes has been recognized as a

580 major challenge in hydrology (Blöschl et al., 2019). Despite our strong awareness of non-stationarity of hydrological parameters, we often lack knowledge to implement system changes in hydrological models. In this proof-of-concept study in the Meuse basin, we propose a top-down approach to introduce a time-dynamic representation of the root-zone storage capacity parameter within process-based hydrological models, using regional climate model simulations. Our approach relies, on the one hand, on a space-for-time exchange of Budyko characteristics of dominant land-use types to estimate the hydrological be-

585 havior of potential land-use changes and, on the other hand, on the interplay between the long-term and seasonal water budgets to represent climate-vegetation interactions under climate and land-use change. Despite knowledge gaps on future ecosystem water use, we implement potential system changes in a hydrological model based on our current understanding of hydrological systems. The predicted hydrological response to 2K warming is strongly altered if we consider that vegetation has adapted its root-zone storage capacity to offset the more pronounced hydro-climatic seasonality under 2K global warming compared to

590 a stationary system. The increased vegetation water demand under global warming results on average annually in -7 % less streamflow, +4 % more evaporation and -6 % less recharge for the scenarios assuming non-stationary conditions compared to a stationary system. These differences even lead to a distinct change of sign of median annual streamflow. Our study contributes to the quest for more plausible representations of catchment properties under change and, therefore, more reliable long-term hydrological predictions.





**Table 1.** Root-zone storage capacity description and symbols, derived from long-term transpiration and storage deficits calculations for observed historical E-OBS data ($P_{\mathrm{obs}}$) and simulated historical ($P_{\mathrm{hist}}$) and 2K climate data ($P_{\mathrm{2K}}$) for historical land use ($\omega_{\mathrm{obs}}$) and land-use change scenarios ($\omega_{\mathrm{broadleaved}}$ and $\omega_{\mathrm{coniferous}}$). The overline symbol is omitted from $P$, $Q$ and $E_{\mathrm{R}}$ to increase readability.

| Description | Root-zone storage capacity $S_{\mathrm{R,max}}$ [mm] | Long-term transpiration $E_{\mathrm{R}}$ [mm yr$^{-1}$] (Eq. 5) | Storage deficit $S_{\mathrm{R,def}}$ [mm] (Eq. 7) |
|---|---|---|---|
| Observed historical E-OBS climate historical land use ($\omega_{\mathrm{obs}}$) | $S_{\mathrm{R,max,A}}$ | $P_{\mathrm{E,obs}} - Q_{\mathrm{obs}}$ | $S_{\mathrm{R,def,obs}}$ |
| Simulated historical climate historical land use ($\omega_{\mathrm{obs}}$) (historical runoff coefficient) | - | $P_{\mathrm{E,hist}} - Q_{\mathrm{obs}}/P_{\mathrm{obs}} \cdot P_{\mathrm{hist}}$ | $S_{\mathrm{R,def,hist}}$ |
| 2K climate historical land use ($\omega_{\mathrm{obs}}$) | $S_{\mathrm{R,max,B}}$ | $P_{\mathrm{E,2K}} - (Q/P)_{\mathrm{2K,B}} \cdot P_{\mathrm{2K}}$ (Eq. 3) | $\max(\lvert S_{\mathrm{R,def,obs}} + \min(0, S_{\mathrm{R,def,2K,B}} - S_{\mathrm{R,def,hist}})\rvert)$ |
| 2K climate broadleaved land use ($\omega_{\mathrm{broadleaved}}$) | $S_{\mathrm{R,max,C}}$ | $P_{\mathrm{E,2K}} - (Q/P)_{\mathrm{2K,C}} \cdot P_{\mathrm{2K}}$ (Eq. 3) | $\max(\lvert S_{\mathrm{R,def,obs}} + \min(0, S_{\mathrm{R,def,2K,C}} - S_{\mathrm{R,def,hist}})\rvert)$ |
| 2K climate coniferous land use ($\omega_{\mathrm{coniferous}}$) | $S_{\mathrm{R,max,D}}$ | $P_{\mathrm{E,2K}} - (Q/P)_{\mathrm{2K,D}} \cdot P_{\mathrm{2K}}$ (Eq. 3) | $\max(\lvert S_{\mathrm{R,def,obs}} + \min(0, S_{\mathrm{R,def,2K,D}} - S_{\mathrm{R,def,hist}})\rvert)$ |





(a)

(b)

Percentage broadleaved forest

25-38%

12-25%

01-12%

(c)

N

50 km

Coniferous forest

Broadleaved forest

Pastures

Cropland

Urban

**Figure 1. (a)** Location of the Meuse basin in North-West Europe. **(b)** Elevation in the basin and categorization of catchments according to their areal percentage of broadleaved forest. **(c)** Main land-use types according to CORINE Land Cover (European Environment Agency, 2018).

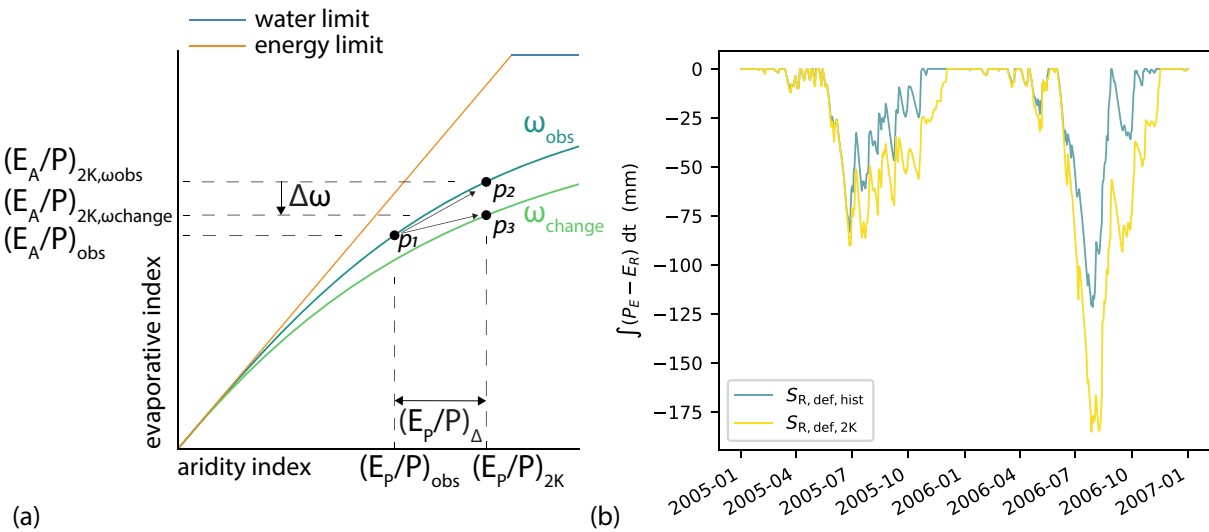

**Figure 2. (a)** Representation of the Budyko space, which shows the evaporative index $(E_A/P)$ as a function of the aridity index $(E_P/P)$ and the water and energy limit. A catchment with aridity index $(E_P/P)_{obs}$ and evaporative index $(E_A/P)_{obs}$, derived from observed historical data, plots at location $p_1$ on the parametric Budyko curve with $\omega_{obs}$. A movement in the Budyko space towards $p_2$ along the $\omega_{obs}$ curve is shown as a result of a change in aridity index $(E_P/P)_\Delta$ towards a projected $(E_A/P)_{2K,\omega obs}$ associated with aridity $(E_P/P)_{2K}$. An additional vertical shift $\Delta\omega$ towards a location $p_3$ on a $\omega_{change}$ curve is shown if additional factors (e.g. land use) are projected to change besides aridity index. Here, the represented downward shift in $\omega$ reduces the change in evaporative index to $(E_A/P)_{2K,\omega change}$. **(b)** Cumulative storage deficits $(S_{R,def})$ derived from effective precipitation $(P_E)$ and transpiration $(E_R)$ using the simulated historical and 2K climate data. Estimates of transpiration $(E_R)$ are derived from long-term water balance projections as a result of movements within the Budyko framework in response to climate and potential land use changes.





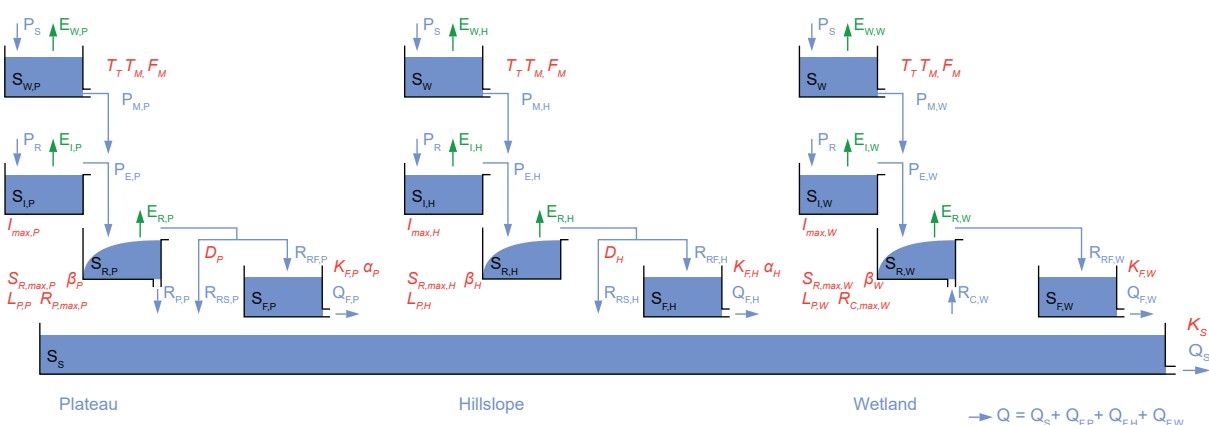

**Figure 3.** Schematic representation of the wflow_FLEX-Topo model with three HRUs for plateau, hillslope and wetland connected through their groundwater storage. The model includes storages for snow $S_W$, interception $S_I$, the root-zone $S_R$, a fast runoff component $S_F$ and groundwater $S_S$ [mm]. The total streamflow $Q$ [mm d$^{-1}$] is the sum of fast runoff $Q_F$ from the three HRUs and groundwater runoff $Q_S$. Evaporation [mm d$^{-1}$] occurs from the snow storage ($E_W$), the interception storage ($E_I$) and the root-zone storage ($E_R$). Main parameters for snow processes include a threshold temperature $T_T$ [$^\circ C$] to distinguish precipitation $P$ falling as rain $P_R$ or snow $P_S$, a threshold temperature for melt $T_M$ [mm d$^{-1}$] and a degree-day factor $F_M$ [mm d$^{-1}$ $^\circ$C$^{-1}$]. For each HRU, other parameters include a maximum interception capacity $I_{max}$ [mm], a maximum root-zone storage capacity $S_{R,max}$ [mm], a shape factor $\beta$ [-], a transpiration water stress factor $L_P$ [-], a factor for the fraction of preferential groundwater recharge $D$ [-], a recession coefficient for the fast storage $K_F$ [d$^{-1}$] and a combined recession for the slow storage $K_S$ [d$^{-1}$]. Parameters specific to plateau, hillslope and wetland include a maximum percolation rate $R_{P,max,P}$ [mm d$^{-1}$], a non-linear coefficient for fast runoff $\alpha_P$ and $\alpha_H$ [-], and a maximum capillary rise rate $R_{C,max,W}$ [mm d$^{-1}$]. Effective precipitation is denoted as $P_E$, fluxes between two stores are denoted as $R$ with subscripts for the stores, and subscripts P, H and W are used to distinguish between the three HRUs.





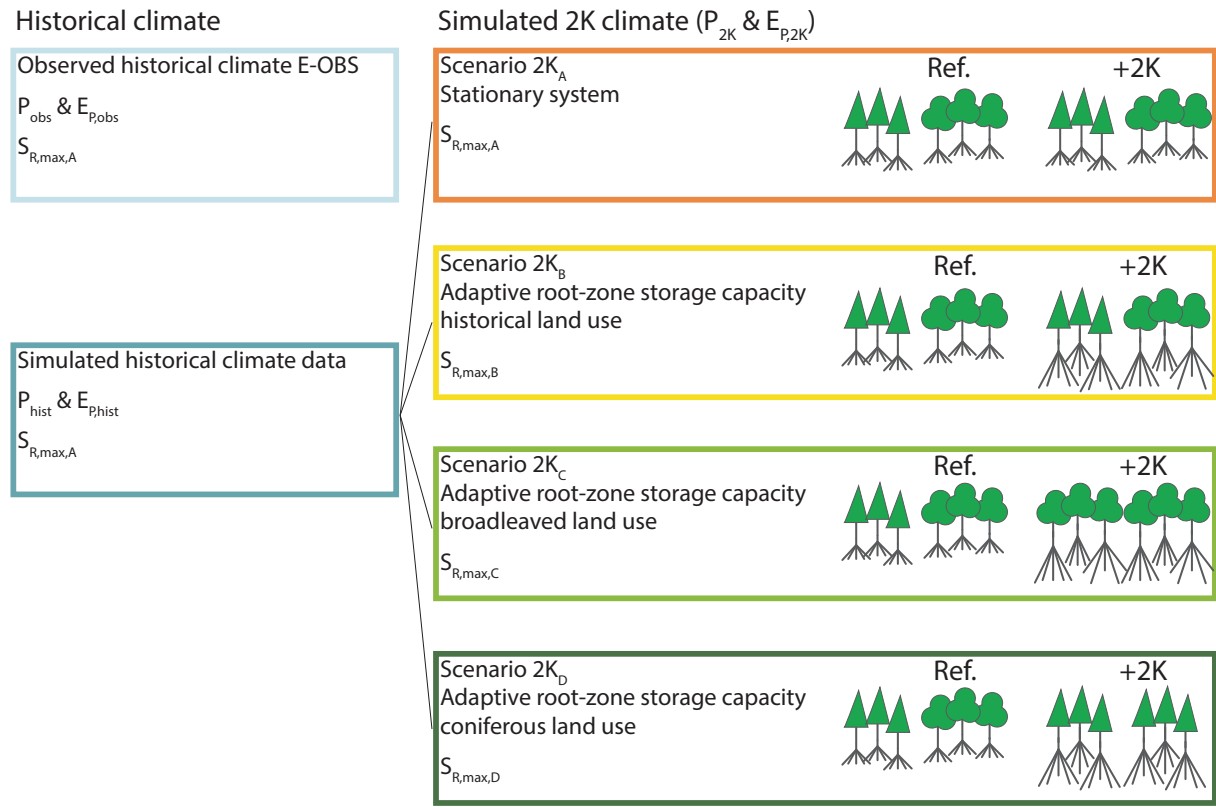

**Figure 4.** Model scenarios using the observed historical and the simulated historical and 2K climate data. The model is calibrated using observed E-OBS data and the historical root-zone storage capacity $S_{R,max,A}$. The model is then forced with the simulated historical climate data using $S_{R,max,A}$ as root-zone storage capacity parameter. We then define four scenarios to compare the change in hydrological response to 2K global warming in comparison to historical conditions for the ensemble of feasible parameter sets. In scenario $2K_A$, we assume an unchanged system (no changes in land use, nor root-zone storage capacity). In scenario $2K_B$, we assume that vegetation has adapted its root-zone storage capacity to the 2K climate, but no changes in land use. In scenario $2K_C$, we test the combination of an adapted root-zone storage capacity in response to the changed climate and a hypothetical conversion of coniferous plantations and agriculture to broadleaved forests in part of the catchment. A similar but reversed approach in land-use changes is assumed in scenario $2K_D$.

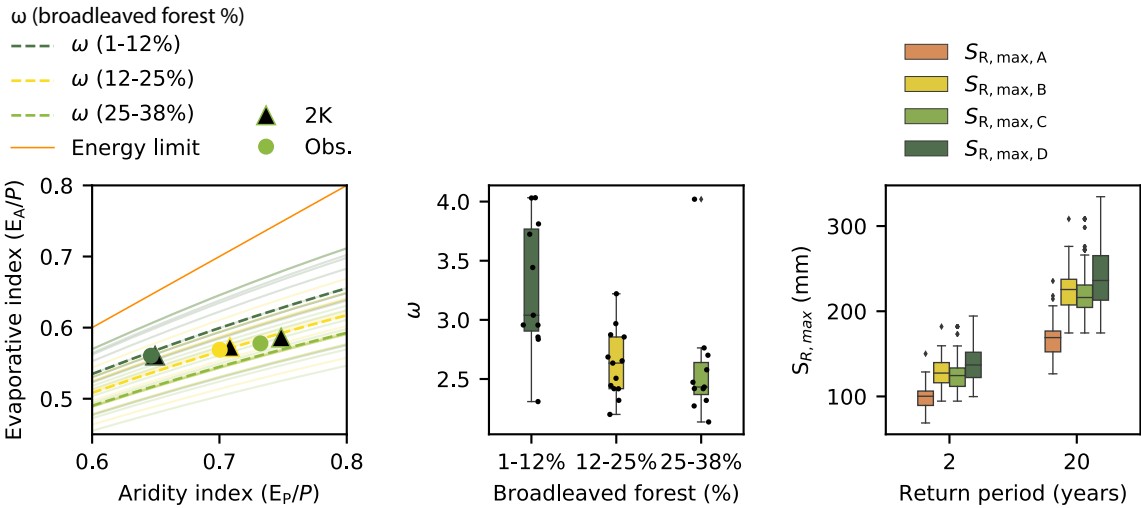

**Figure 5. (a)** Budyko space with parametric $\omega_{obs}$ curves for each of the 35 catchments of the Meuse basin, categorized according to their percentage of broadleaved forest. The dashes curves represent the median $\omega_{obs}$ curves for each category. The change in aridity index from historical to 2K climate conditions along each parameterized $\omega_{obs}$ curve is also shown for the median of the three categories. **(b)** Parameterized $\omega_{obs}$ values for each of the 35 catchments of the Meuse basin, categorized according to their percentage of broadleaved forest. **(c)** Range of root-zone storage capacities across the 35 catchments of the Meuse basin for the four scenarios. $S_{R,max,A}$ represents the root-zone storage capacity for historical conditions. $S_{R,max,B}$ represents an adapted root-zone storage capacity in response to the 2K climate but no land use change. In the estimation of $S_{R,max,C}$, catchments with a low percentage of broadleaved forest (1-12%) receive $\omega$ values sampled from catchments with a high percentage of broadleaved forest (25-38%), to represent changes in land use towards a conversion to broadleaved forest. A similar but reversed approach is applied for the estimation of $S_{R,max,D}$.

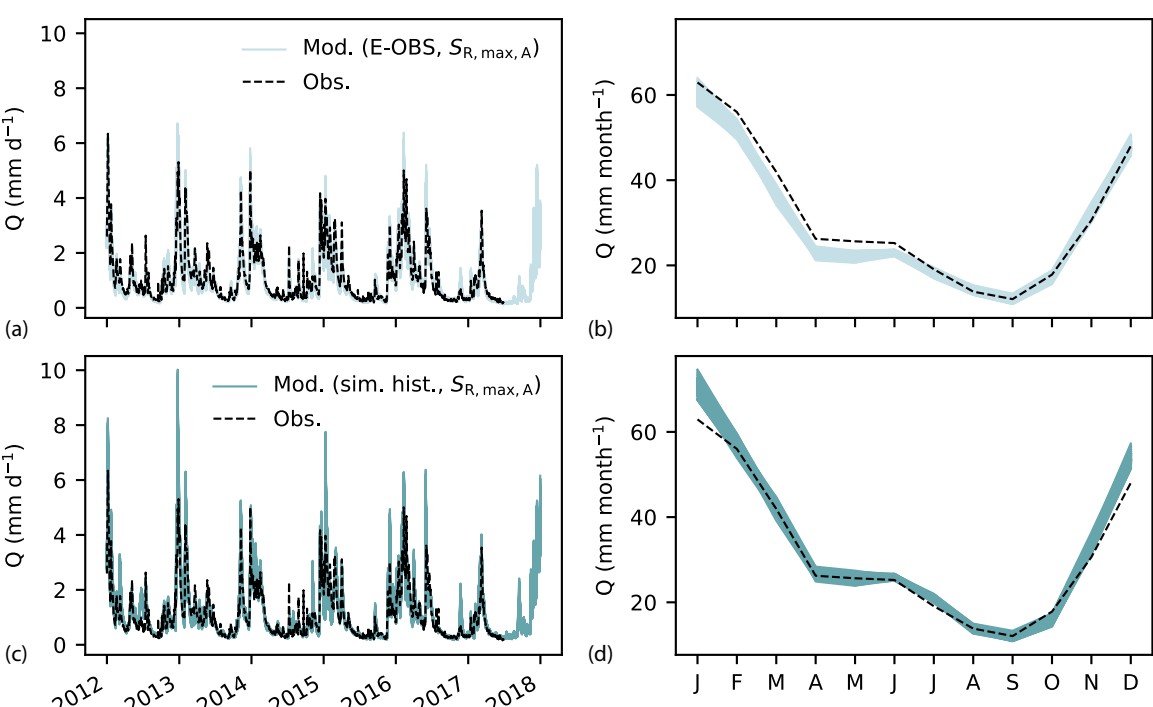

**Figure 6.** Observed and modeled hydrographs and mean monthly streamflow at Borgharen for the ensemble of parameter sets retained as feasible after calibration when the model is: **(a,b)** forced with E-OBS historical data and using $S_{R,max,A}$ as model parameter, and **(c,d)** forced with the simulated historical climate data using $S_{R,max,A}$ as model parameter.



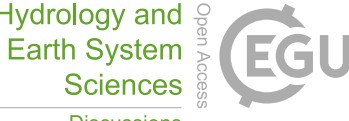

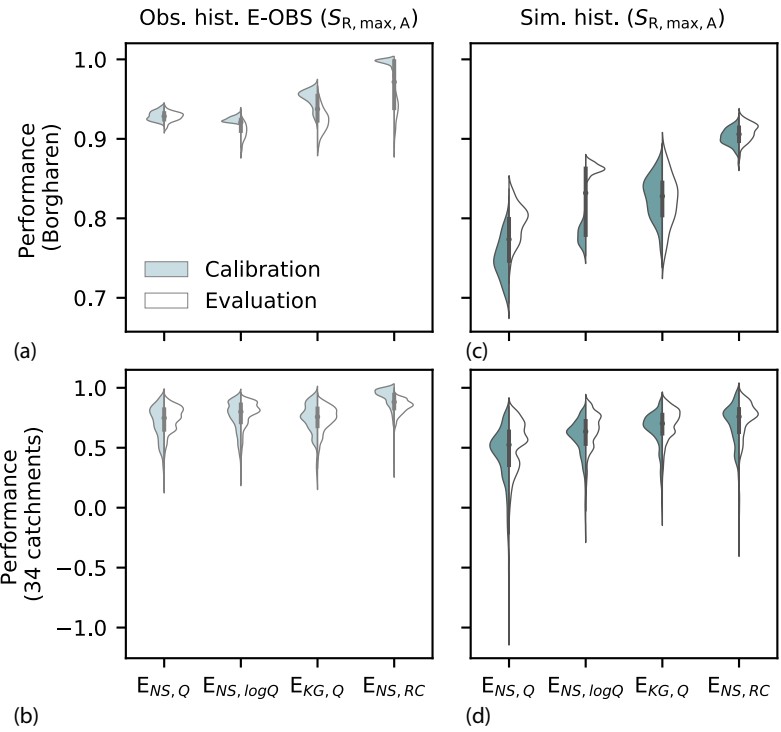

**Figure 7.** Streamflow model performance during calibration and evaluation for the four objective functions when the model is forced with (**a,b**) observed historical E-OBS data and (**c,d**) simulated historical climate data at (**a,c**) Borgharen and (**b,d**) for the ensemble of nested catchments in the Meuse basin. The four objective functions are the Nash-Sutcliffe efficiencies of streamflow, logarithm of streamflow and monthly runoff coefficient ($E_{NS,Q}$, $E_{NS,logQ}$, $E_{NS,RC}$) as well as the Kling-Gupta efficiency of streamflow ($E_{KG,Q}$). Note the different y-axis between rows.

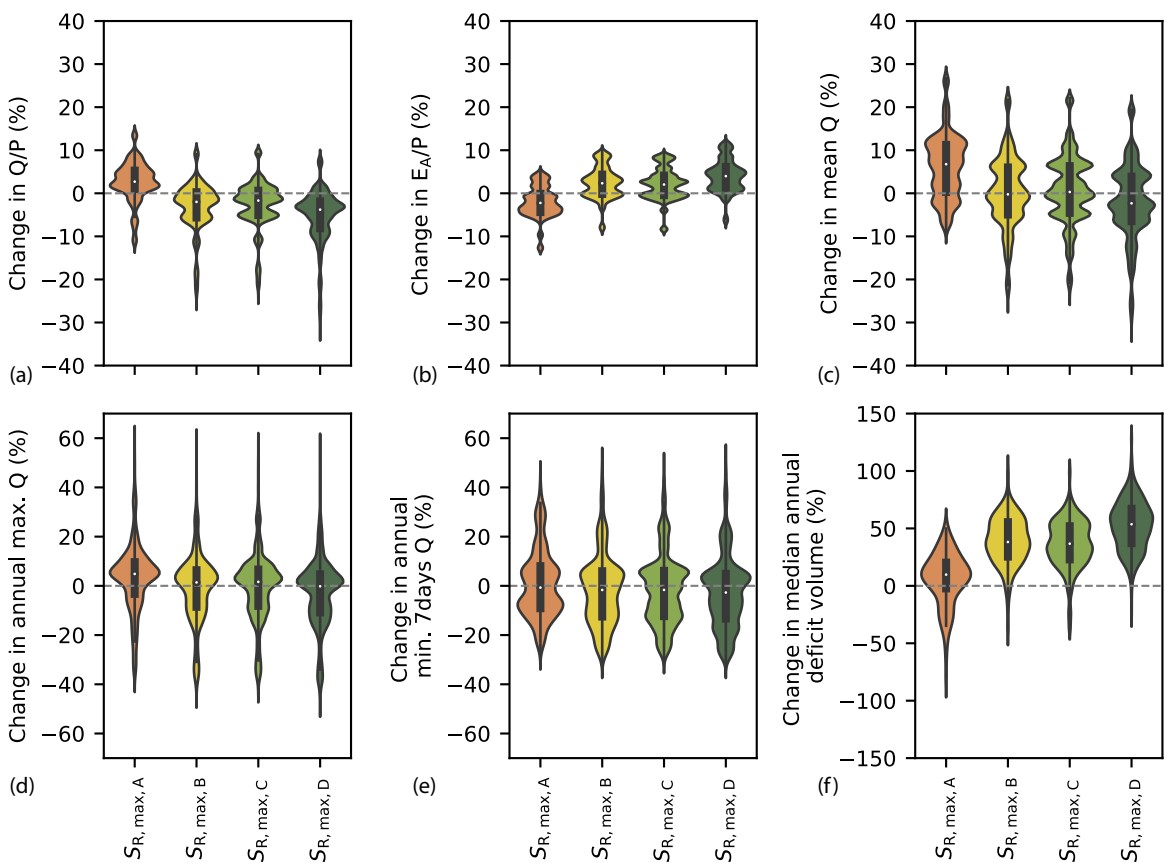

**Figure 8.** Percentage change in annual hydrological response indicators between the 2K and historical model runs for the four scenarios, each based on different assumptions for the root-zone storage capacity parameter $S_{\mathrm{R,max}}$. Percentage change in **(a)** runoff coefficient $Q/P$, **(b)** evaporative index $E_{\mathrm{A}}/P$, **(c)** mean annual streamflow, **(d)** mean annual maximum streamflow, **(e)** minimum annual 7-days mean streamflow, **(f)** median annual deficit volume below the reference 90$^{\mathrm{th}}$ percentile streamflow.



**Figure 9.** Percentage change in mean monthly hydrological response of several flux and state variables between the 2K and historical model runs for the four scenarios, each based on different assumptions for the root-zone storage capacity parameter $S_{\mathrm{R,max}}$. Percentage change in mean monthly **(a)** streamflow $Q$, **(b)** actual evaporation $E_{\mathrm{A}}$, **(c)** root-zone storage $S_{\mathrm{R}}$, **(d)** groundwater storage $S_{\mathrm{S}}$.



## 8  Data availability

Streamflow data for the Belgian stations are provided by the Service Public de Wallonie in Belgium (Direction générale opérationnelle de la Mobilité et des Voies hydrauliques, Département des Etudes et de l'Appui à la Gestion, Direction de la Gestion hydrologique intégrée (Bld du Nord 8-5000 Namur, Belgium)). Streamflow data for the French stations are retrieved from the Banque Hydro portal (http://www.hydro.eaufrance.fr/). The E-OBS dataset (v20.0e) for daily precipitation, temperature and radiation fields for the historical period is used (Cornes et al., 2018) and can be downloaded from https://www.ecad.eu/download/ensembles/download.php. The simulated historical and 2K climate data are provided by the Royal Netherlands Meteorological Institute (KNMI).

*Author contributions.* HHG, MH, EEA and LJEB designed the study. EEA provided the simulated historical and 2K climate data. LJEB conducted all the analyses and wrote the manuscript. All authors discussed the results and contributed to the final manuscript.

*Competing interests.* The authors declare that they have no competing interests.

*Acknowledgements.* We thank Deltares and Rijkswaterstaat for the financial support to conduct this analysis. The authors would like to thank the Service Public de Wallonie, Direction générale opérationnelle de la Mobilité et des Voies hydrauliques, Département des Etudes et de l'Appui à la Gestion, Direction de la Gestion hydrologique intégrée (Bld du Nord 8-5000 Namur, Belgium) for providing the streamflow data. We thank Erik van Meijgaard for performing the historical and 2K climate data simulations. We also thank Wouter Berghuijs for his valuable advice on the Budyko framework.





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
