# Peer review of "Ecosystem adaptation to climate change: the sensitivity of hydrological predictions to time-dynamic model parameters"

_Hydrology and Earth System Sciences, 2021_

## Referee Comment (RC1)

**Review of the manuscript "The importance of ecosystem adaptation on hydrological model predictions in response to climate change", ID hess-2021-204**

**General comments**

As my background lies in ecological and ecohydrological modelling, my ability to judge the novelty of this study in the field of hydrological modelling is limited. I rather take the perspective of the interested ecologist, who appreciates this important step to integrate vegetation in a more realistic way into a hydrological model.

**Short summary**

In this proof-of-concept study, the authors propose a top-down approach to include potential land-use and vegetation changes under climate change in process-based hydrological models to make predictions of future hydrological conditions more accurate.

In a multistep process, the root-zone storage capacity is calculated for different climate and land-use scenarios using observed climate, streamflow and land cover data and simulated historical and 2K climate change data from the Meuse Basin.

The model is calibrated and validated for historical climate and land-use data and run with 4 different climate change scenarios (all 2K warming): a stationary system (no change in root-zone storage capacity and land-use), a system with no land-use change but an adaptation of the root-zone storage capacity to changed climate and, a system with adaptation of the root-zone storage capacity to changed climate combined with land-use change from broadleaved forest to agriculture/coniferous forest and vice versa.

The results show that root-zone storage capacity parameter increased with climate change whereas it differs only slightly between the different land-use scenarios.

The catchment hydrological response of the stationary root-zone storage scenario differed from the scenarios with adapted root-zone storage capacity. The non-stationary systems showed higher evaporation, lower runoff coefficients and streamflow. Land-use change did not influence the results to the same extent as adaption of the root-zone storage capacity.

In the discussion, the authors emphasize the impact of vegetation (root-zone storage capacity) on the water cycle and propose their approach as a way to move past stationary hydrological systems when modelling climate change impact on hydrology with process-based hydrological models.

**Scope and conclusions**

Overall, the manuscript is well written and structured and the method provides an interesting solution to adapt the hydrological system to climate change by using available data on a catchment. The stepwise approach allows to systematically assess the effects of land-use and climate change separately and combined.

The manuscript addresses relevant scientific questions of the impact of land-use and climate change on ecosystem properties with relevance for hydrological models. This is an important question and will contribute to an improved prediction of climate change impacts on hydrological processes in the future. The questions addressed in the manuscript lie within the scope of the HESS journal that publishes research on the role of biological processes on continental water cycles, temporal characteristics of water resources and the impact of human activity on the water cycle.

**Main concerns**

**Study objectives are not clearly and consistently stated**

According to the abstract, introduction and conclusion, the study has two main objectives. The first is to propose a top-down approach to include vegetation change into hydrological models via the root-zone storage capacity (l. 4-5, 575, 581-583). The second is the quantification of the sensitivity of modelled hydrology to changes in root-zone storage capacity under climate change and related to that, the testing of the hypothesis that changes will be more pronounced when considering an adapted root-zone storage (l. 93 ff).

Although these two objectives are clearly connected, they are never stated together. The first objective (proof-of-concept and methodological aspect) of the study is stressed in the discussion and conclusion, whereas the introduction highlights only the second objective (application and sensitivity analysis). The objectives of the study should be more clearly stated in the introduction and the discussion and conclusion should build on these objectives.

**Discussion and conclusions leave open questions**

The discussion could be more thorough and consistent regarding both, the modelling results and the methodological approach.

The discussion is structured into two separate parts: *Implications* (l. 500- 539) and *Limitations and knowledge gaps* (541-577).

However, two paragraphs from the first section (Implications) are better suited for the second section (limitations): l. 512-519 on possible further exploration of the space-for-time concept and l. 535-539 on the limitations of the simulated climate time series used in the study.

Also, given that one major objective of the study is to propose an approach to include vegetation change into hydrological models, I feel that the model results are not thoroughly discussed as to whether the proof-of concept of the method was successful.

The following questions/issues remain unaddressed:
  - The approach showed that the root-zone storage capacity parameter has a potentially large effect on future water flows. How realistic are the values for root-zone storage capacity that were calculated for the different scenarios? Is there any evidence from literature regarding the extent to which plants adapt their root system to changing climate? Does this adaptation depend on vegetation type (e.g. crop/grass vs. tree) or species?
  - Are the results regarding the water flow under future conditions realistic? Is this what could be expected under climate change?
  - In which situations can this method be applied? Which hydrological models? Which ecosystems?
  - What are the limitations and chances of this approach?

**Methods: no limitation of the root-zone storage capacity**

The methodological approach assumes a limitless adaptation of the root-zone storage capacity to changing aridity index (compare l. 243). I was wondering whether this is realistic. The adaptability of the root-zone depends on the vegetation's capability to change the root system following a change in climate/water demand. This capability probably depends on the vegetation type (crop, grass, or tree) but also on the species. Also, adapting the root-zone storage capacity is not the only way that plants/vegetation might adapt to a change in aridity index. Plants can adapt to drier conditions by closing their stomata and reducing gas exchange with the atmosphere and hence transpiration. Also, overall vegetation cover could decrease if the water supply is not sufficient to support the same cover. Although I think it is not necessary to consider this limitation in this proof-of concept study, it is nevertheless an important point to discuss in the discussion section.

**Links to ecohydrological modelling or dynamic global vegetation models (DGVMs) missing**
Although this study is about hydrological modelling, I think that the advances and contributions of ecohydrological models and DGVMs to studying the feedbacks between vegetation and the water cycle should be mentioned and discussed in the introduction and, if applicable, also in the discussion section of the manuscript. Please find some hints on where to start in the following:

One prominent model is e.g. the DGVM LPJmL which dynamically models carbon, nitrogen and water flows. The model has been applied to various question among them also questions related to water flows under climate and land-use change.

You can e.g. have a look at the following publication:
Rost et al. (2008), *Water Resources Research*. https://doi.org/10.1029/2007WR006331

Here you can find a list of some key publications of the model:

https://www.pik-potsdam.de/en/institute/departments/activities/biosphere-water-modelling/lpjml/key-publications
In the field of ecohydrological modelling, you could have a look at the works of Ignacio Rodriguez-Iturbe and Amilcare Porporato. An ecohydrological study to look at might be Tietjen et al. (2017), *Global Change Biology* https://doi.org/10.1111/gcb.13598. The study looks at feedbacks between soil water availability, vegetation change and climate change and they disentangle the effects of climate change alone and climate change in combination with vegetation change.

**Specific comments**
**Introduction**

l. 58: optimality principles: is this an established term? If not specify what is optimized in this approach (probably it's vegetation growth or something similar)

l. 95: "land-use change under future conditions": The manuscript does not tackle land-use under future conditions. The authors test what happens if land-use is the same in the whole catchment based on what is already there. But it is never discussed which land-use types are realistic for the future or whether there is a trend in land-use towards any of the present land-use types. Rephrase to make clear that this is just a theoretical assessment of the sensitivity towards different types of land-use instead of a projection into the future. Also, the statement "we exchange space-for-time" (l. 96) suggests, that there is a known land-use trend for the future.

**Methods**

**General comment**
The method description is generally a bit confusing. I feel that generally it could be a bit shorter (e.g. the scenario description and the description of the 4 different root-zone storage capacities) are repetitive at some points. It might also help to provide a supportive figure of the study's workflow that clearly separates between different sources of input data, generation of scenarios and model application (instead of Fig. 3 which would fit better in the Supplemental material). Please revise the method section for more clarity and structure. The specific comments below hopefully help to do that.

**Study area**

l. 109: "divided into three main zones": It would be nice to see these three main zones in the Figure as well. In the figure, it is unclear which part of the catchment represents which of these three zones.

l. 120: reference is missing for the meteorological variables

l. 122: always refer to the specific label of the figure if possible (here it's Fig. 1c and not Fig. 1)

**Data**

l. 147-161: A figure or some numbers comparing the simulated historical and 2K climate scenarios could be a nice addition. From the description, it remains unclear what a "globally 2K warmer world" (l. 158) will translate to in this regional data set. Does this 2K warmer world lead to a mean 2K warmer regional climate? What's the difference in mean annual temperature and mean annual precipitation in 2K vs. historical climate?

l. 164: It would be helpful to add Borgharen to the catchment map in Fig. 1

**Methods**

General: The decision to divide the land-use types into broadleaved forest on the one hand, and coniferous forest/agriculture on the other hand needs better explaining. Why is a tree-dominated (coniferous) vegetation grouped with crops? I would expect that crops and trees are very different with regard to their effect on the water cycle and concerning their root-storage capacity.

l. 233: Why is Imax taken as 2mm?

l. 262: Why are E-OBS data taken from 1980-2018 while streamflow data is only from 2005-2017? Would the results have been different if E-OBS data from 2005-2017 were used instead?

l. 289 ff: How were the $\omega$ values sampled?

l. 306: hillslopes are associated with forest and plateau with agriculture. But which type of forest do you mean here? Broadleaved or coniferous?

l. 331ff: "the performance … for the ensemble of retained parameter sets": From the 10000 calibration runs: how many parameter sets were obtained for the model runs? From the supplemental material it looks like the prior is almost the same as the posterior parameter distribution.

l. 334-337: This section can be removed as it is a repetition of what was already mentioned above in lines 272-274.

Scenario description in 4.4:

   - It is unclear which values of $S_R$ with regard to the return period are used (2 years or 20 years?)
   - How did you decide for the return period in the mixed agricultural/coniferous land-use? Agriculture should be 2 years and forest 20 years (l. 251-253)

l. 357, 362, 369: no need to repeat that $S_{Rmax,a}$ is used as a parameter in the historical run for every scenario. Better to mention it once, when the historical run is explained.

**Results**

General comment: It is not always clear what the reported numbers represent. Median and standard deviation? Mean and standard error of the mean? E.g. l. 374 & 377, l. 382, l. 390 & 391, l. 402, l. 408. If the reported values are always the same, you could also mention it once and state that all subsequent values represent the same measures.

l. 376: should this be $\omega_{obs}$ instead of $\omega$?

l. 377: should this be transpiration instead of evaporation? This is a general issue: there is no clear distinction between evaporation, transpiration and evapotranspiration in the text.

l. 377-379: The differences of $\omega$ between the catchments is mainly attributed to the differences in the main vegetation type (broadleaved vs. coniferous/agriculture, l. 377-379). However, the catchments also differ substantially in other characteristics (French part: thick soils and gentle slopes, thin soils and steep terrain in the Ardennes, porous chalk in Wallonia (l. 109-113)). It should be discussed to what extent the differences in $\omega$ might not be dependent on the vegetation cover alone but also on

the topography and soil type/thickness. Also, what are the implications of this regarding the method? How sure are you that the differences in hydrology between land-use types are really caused by the vegetation cover and not by the underlying topographical and soil characteristics?

l. 394: Fig. 2b should either be referenced earlier in the text, e.g. when talking about the difference between the historical and the 2K climate time series in the method section or it should be a separate result figure that comes later in the text.

l. 424: "median values of approximately 0.93": why approximately?

l. 431: "streamflow during the wettest months": include which months you mean by "wettest months"

Results for ω depending on the

**Discussion**

**Implications**

l. 500: "shows distinct patterns of change": more precise language could be used: Which response variables differ and are they larger or smaller compared to the stationary scenario?

l. 512-519: This section does not fit in the "Implications" section of the discussion. It is more of a limitation of the current study or an outlook of what could be done next. It could e.g. be moved to the "Limitations and Knowledge gaps" section of the discussion.

l. 524-256: It is not clear to me, why the results on actual evaporation differences between the scenarios indicate disagreements among model process representations. Please elaborate more on this point. Also, what are the specific "processes that become relevant in the future"?

l. 333-334: The conclusion, that vegetation is important for regulating the water cycle is correct but it is also quite established and not really a specific discussion of your results.

l. 535-539: This discussion is also a limitation of your study or an outlook to further work. It should not be under the "Implications" subheading of your discussion.

**Limitations and knowledge gaps**

l. 542: "it is unclear how ecosystems will cope with climate change": A discussion of how useful your approach to include vegetation into hydrological models under climate change in the light of this uncertainty would be interesting. To what extent can we be sure that the root-zone storage capacity can adapt to changing climate? What evidence is there from other studies regarding this issue? How would you proceed with your approach if vegetation changes to a vegetation type for which there is no data from the same region?

At the end of the discussion, you mention several times that this study should be read as a sensitivity analysis (l. 571) and a proof-of-concept (l. 575). This should also be made clear in the abstract. Also, a thorough discussion of the advantages and disadvantages of the presented method is missing. What are possible applications of it, to what types of regions/questions can it be applied? What are the limitation and what could be improved?

**Figures**
- General: figure labels should be in the same position for all figures (e.g. top left)
- Figure labels could be bold for better visibility?
- Why are the scenario names (2Ka-d) that are defined in Fig. 4 never used? Instead $S_{rmaxa-d}$ is used in Figs. 5,8,9? If scenario names are given, they should be used consistently.

**Fig. 1:**
- colours of figure b): better use some continuous colour scheme

- Figure labels are inconsistent, b and c not on the same height
- Fig. 1b: what are the black points? Are they the streamflow measurement locations? Mention in the caption
- Fig. 1 does not reflect well many aspects mentioned in the text (2.1 landscape and 2.2 land use)
  - Which are the three zones mentioned in l. 109? Are they represented in Fig. 1b? If yes you could add this to the caption. It is not clear what is the French, the Ardennes and the Wallonia part mentioned several times in the text
  - Fig. 1b: The numbers don't really match with the text. In Walloon 44% of the broadleaved forest should be there (l. 126), but in the figure the max. percentage is 38%.

**Fig. 3:**
- Maybe this figure fits better in Supplement S3 because it is part of the model description? I don't find it very helpful in the manuscript without the context of the model formulas

**Fig. 5:**
- Labels are missing
- Figures are a bit small: Could be a made bigger if empty space between panels is reduced
- 5b:
  - $\omega_{obs}$ should be on the y-axis not just $\omega$
  - Axis text: No % because it's already in x-axis title
- 5c:
  - Caption last sentence: "A similar but reversed approach is applied …" It is the *same* and not a *similar* approach that was used.

**Fig. 6:**
- What is the ribbon for the modelled values: range from all realistic parameter sets of the calibration?

**Fig. 7:**
- Could be larger: box is not visible
- Don't use transparent colours to distinguish the panels. In my opinion they are already distinguished enough by the panel titles and labels in the caption (same for figures in Supplement S3)
- Labelling is not consistent (compare to labelling of Fig. 6)
- Why is there such a big difference between Borgharen and the 34 catchments? Isn't Borgharen just a summary of all the catchments?

**Fig. 8:**
- Caption 8e) maybe mention that y-axis is different scale (compare to caption of Fig. 7)

**Fig. 9:**
- What are the ribbons and lines? Median + conf. interval?

**Supplemental Material**
**S1: Monthly correction factors for E-OBS precipitation data**
- First sentence: Citation missing

**S4: Prior and posterior parameter distributions**
- State in table heading, that the last 3 columns are the posterior parameter distributions

**Technical corrections**

l. 54: rephrase to: sensitivity of the hydrological response to change in …

l. 62: remove "as often referred to"

l. 79: remove the full stop before the list of references

l. 191 & 1197: same style for (p1), (p2) and p3 (either with or without brackets)

l. 392: replace "return periods of 2 year" with either "2 year return period" or "return period of 2 years". Also check the subsequent text as this mistake happens several times.

l. 410: Vertical space is missing as a new paragraph begins in line 411

l. 500: "compared to" instead of "with respect to"?

l. 592: "distinct change of sign": remove distinct

Avoid unspecific adverbs. Either remove them, or state specifically what you mean by them. E.g.

- l.114: "relatively short response time" (how short is relatively short?)
- l. 422: "relatively well reproduced"
- l.423: "slight underestimation" and "relatively similar performance"

---

## Author Comment (AC1)

*We thank the anonymous referee for his/her positive, thorough and constructive review. We provide an answer to each comment below.*

**Comment 1:**

**Study objectives are not clearly and consistently stated**
According to the abstract, introduction and conclusion, the study has two main objectives. The first is to propose a top-down approach to include vegetation change into hydrological models via the root-zone storage capacity (l. 4-5, 575, 581-583). The second is the quantification of the sensitivity of modelled hydrology to changes in root-zone storage capacity under climate change and related to that, the testing of the hypothesis that changes will be more pronounced when considering an adapted root-zone storage (l. 93 ff).
Although these two objectives are clearly connected, they are never stated together. The first objective (proof-of-concept and methodological aspect) of the study is stressed in the discussion and conclusion, whereas the introduction highlights only the second objective (application and sensitivity analysis). The objectives of the study should be more clearly stated in the introduction and the discussion and conclusion should build on these objectives.

**Reply 1:**

*We agree that these are two main aspects of the manuscript. However, in the revised version of the manuscript, we will more clearly state that our objective is to evaluate the sensitivity of hydrological model predictions to ecosystem adaptation in response to climate and potential land use change. To reach this objective, we introduce an approach, subject to assumptions, to estimate future evaporation and associated changes in the root-zone storage capacity. As the future is unknown, we cannot evaluate our results against observations. Therefore, we would rather not qualify the introduced methodology as an objective as the underlying hypothesis cannot be tested. In the revised version of the manuscript, we will not use the terminology "proof-of-concept". However, we fully agree with the suggestions of the reviewer to more thoroughly discuss the limitations and opportunities of the proposed methodology in the discussion (see our reply to the next comment).*

**Comment 2:**

**Discussion and conclusions leave open questions**
The discussion could be more thorough and consistent regarding both, the modelling results and the methodological approach.
The discussion is structured into two separate parts: *Implications* (l. 500- 539) and *Limitations and knowledge gaps* (541-577).
However, two paragraphs from the first section (Implications) are better suited for the second section (limitations): l. 512-519 on possible further exploration of the space-for-time concept and l. 535-539 on the limitations of the simulated climate time series used in the study.

Also, given that one major objective of the study is to propose an approach to include vegetation change into hydrological models, I feel that the model results are not thoroughly discussed as to whether the proof-of concept of the method was successful.
The following questions/issues remain unaddressed:
   a)   The approach showed that the root-zone storage capacity parameter has a potentially large effect on future water flows. How realistic are the values for root-zone storage capacity

that were calculated for the different scenarios? Is there any evidence from literature regarding the extent to which plants adapt their root system to changing climate? Does this adaptationdepend on vegetation type (e.g. crop/grass vs. tree) or species?

b) Are the results regarding the water flow under future conditions realistic? Is this what couldbe expected under climate change?

c) In which situations can this method be applied? Which hydrological models? Which ecosystems?

d) What are the limitations and chances of this approach?

**Reply 2:**

*In the revised version of the manuscript, we will more thoroughly and consistently discuss the limitations and opportunities of the methodological approach and the modelling results.*

*We agree that the two paragraphs in the Implication section mentioned by the reviewer are more related to "Outlooks" of possible future work. Initially we had treated this as implications of our work for future work, but we agree that it may better fit in a section called: "Limitations and outlook". We will adapt this in the revised version of the manuscript.*

*As the future is unknown, we cannot evaluate our results against observations. Therefore, "proof-of-concept" may not be the right terminology, we will adapt this in the revised version of the manuscript. Our study should really be considered as a sensitivity analysis to test the sensitivity of hydrological model predictions to non-stationary systems through plausible assumptions of ecosystem adaptation. To emphasize this more strongly, we propose to adapt the title of the manuscript to: "The sensitivity of hydrological model predictions to ecosystem adaptation in response to climate change."*

*In the discussion of the revised manuscript, we will address the following discussion points:*

a) *The estimated values of the root-zone storage capacity for the different scenarios have median values below 250 mm for a return period of 20 years, which is within the range of global root-zone storage capacity values estimated by Wang-Erlandsson et al. (2016). We will mention this in the discussion of the revised manuscript.*

*There is increasing evidence that vegetation efficiently adapts to its (changing) environment (Gentine et al., 2012, Troch et al., 2013, Hrachowitz et al., 2020). Guswa (2008) shows that the active root zone tends to be larger in water-limited ecosystems in comparison to wet environments. A distinction should be made between individual plant adaptions of roots and the adaptation of the root system of the collective of plants at the ecosystem scale. The study of Brunner et al. (2015) describes several strategies of tree root to cope with drought, which include root biomass adjustments, anatomical alterations and physiological acclimations. Individual plants that have not adapted to meet their water and light requirements will disappear and be replaced by better adapted plants. Therefore, the root system at the ecosystem scale and associated root-zone storage capacity continuously adapt to changing environmental conditions in a state of dynamic equilibrium (Hrachowitz et al., 2020). While the adaptation of individual plants depends on vegetation type and species, here, we determine effective values of the root-zone storage capacity at the catchment scale to reflect the adaptation of the whole ecosystem.*

b) *Given the changes in temperature and precipitation, the future predicted hydrological response does not seem unrealistic, although, of course, this cannot be tested against observations. Common practice in hydrological studies on the impact of climate change is to assume a stationary system (Benchmark Scenario $2K_A$ in our analysis). In addition to this*

*scenario, we suggest a possible approach to consider ecosystem adaptation in response to climate change and test the sensitivity in the resulting hydrological response. Our approach is subject to considerable uncertainties in the estimation of the future transpiration (required to estimate the root-zone storage capacity) as we are using the Budyko framework for future conditions (Berghuijs et al., 2020; Reaver et al. 2021). Besides, we do not explicitly consider that vegetation can adapt to drier conditions by regulating their stomata and hence reducing transpiration (which is the topic of your comment 3). Moreover, the increased $CO_2$ concentration may, on the one hand, increase water use efficiency, while on the other hand increase green foliage due to fertilization effects (Donohue et al., 2013; Frank et al., 2015; Yang et al., 2019). Hence, we cannot predict what will exactly happen, but we can at least test the sensitivity of the hydrological response to changes in the system representation.*

c)  *Root-zone storage capacity estimates derived from the water-balance approach are applicable in various hydrological and land surface models, provided that they include a root-zone parameterization, which is the case for most models (Nijzink et al. 2016, van Oorschot et al., 2021). The water-balance approach to estimate the root-zone storage capacity has successfully been applied in a variety of climate zones and across various ecosystems (New-Zealand in de Boer et al. 2016; Australia in Donohue et al., 2012, United States in Gentine et al. 2012 and Gao et al., 2014; and at the global scale in Wang-Erlandsson et al., 2016). The method was also applied along rainforest-savanna transitions to reveal drought-coping strategies (Singh et al., 2020). However, the method is not suitable in areas where the water table is very close to the surface and where vegetation directly can tap from the available groundwater instead of creating a buffer capacity (e.g. Fan et al. 2017). Another limitation of the water-balance approach relates to equation 6, in which we scale the daily transpiration estimates with a constant factor to the patterns of potential evaporation minus interception evaporation, implying that vegetation can extract water for transpiration from dry soils as easily as from wet soils.*

d)  *The proposed methodology to estimate future root-zone storage capacities relies on the underlying assumption that past empirical relations between aridity index and evaporative index (i.e. the Budyko framework) still apply in the future. The Budyko framework reflects the long-term hydrological partitioning under dynamic equilibrium conditions. Therefore, when using the Budyko framework to estimate the future rate of transpiration, we assume that the future vegetation has adapted to the future climatic conditions and that it is in a state of dynamic equilibrium. This is a considerable uncertainty of our methodology because it implies that vegetation has had the time to adapt to the rapidly changing environmental conditions. There is no doubt that vegetation eventually will adapt, otherwise we would not see the hydrological partitioning of catchments around the world broadly plotting along the Budyko curve. However, unanswered questions are how long it will take for vegetation to adapt and how it will adapt. While the Budyko framework is a well-established concept, the recent study by Reaver et al. (2021) shows that it should be cautiously applied in changing systems which are not in equilibrium. We will include this discussion in the revised version of the manuscript.*

*Despite these uncertainties, there are also strong aspects of our methodology. Current practice in most climate change assessment studies assumes constant system properties in the future, thereby neglecting adaption of vegetation to local climate conditions. Our analysis is a first step in evaluating what may happen if we consider ecosystem adaptation in response to climate change in hydrological model predictions. Our method is based on readily available data and is therefore easily applicable. Furthermore, if we assume space and time symmetry, i.e. the exchange of spatial knowledge with temporal knowledge, we may be able to transfer root-zone storage capacity estimates from a location X with a current climate similar to the future climate of a location Y.*

**Comment 3:**

**Methods: no limitation of the root-zone storage capacity**
The methodological approach assumes a limitless adaptation of the root-zone storage capacity to changing aridity index (compare l. 243). I was wondering whether this is realistic. The adaptability of the root-zone depends on the vegetation's capability to change the root system following a change inclimate/water demand. This capability probably depends on the vegetation type (crop, grass, or tree)but also on the species. Also, adapting the root-zone storage capacity is not the only way that plants/vegetation might adapt to a change in aridity index. Plants can adapt to drier conditions by closing their stomata and reducing gas exchange with the atmosphere and hence transpiration. Also,overall vegetation cover could decrease if the water supply is not sufficient to support the same cover.Although I think it is not necessary to consider this limitation in this proof-of concept study, it is nevertheless an important point to discuss in the discussion section.

*Reply 3:*

*This is a very good point, we briefly mention it in the discussion when we refer to the study of Zhang et al. (2020). However, we fully agree that the different strategies of vegetation to cope with changing environmental conditions need to be discussed in more detail in the revised version of the manuscript.*

**Comment 4:**

**Links to ecohydrological modelling or dynamic global vegetation models (DGVMs) missing** Although this study is about hydrological modelling, I think that the advances and contributions ofecohydrological models and DGVMs to studying the feedbacks between vegetation and the watercycle should be mentioned and discussed in the introduction and, if applicable, also in the discussionsection of the manuscript. Please find some hints on where to start in the following:

One prominent model is e.g. the DGVM LPJmL which dynamically models carbon, nitrogen and waterflows. The model has been applied to various question among them also questions related to water flows under climate and land-use change.

You can e.g. have a look at the following publication:
Rost et al. (2008), *Water Resources Research*.

https://doi.org/10.1029/2007WR006331Here you can find a list of some key

publications of the model:

https://www.pik-potsdam.de/en/institute/departments/activities/biosphere-water- modelling/lpjml/key-publications
In the field of ecohydrological modelling, you could have a look at the works of Ignacio Rodriguez-Iturbe and Amilcare Porporato. An ecohydrological study to look at might be Tietjen et al. (2017), *Global Change Biology* https://doi.org/10.1111/gcb.13598. The study looks at feedbacks betweensoil water availability, vegetation change and climate change and they disentangle the effects of climate change alone and climate change in combination with vegetation change.

*Reply 4:*

*We thank the reviewer for providing the references of these relevant studies on ecohydrological modelling. It is interesting to read that in the study of Tietjen et al. (2017), the future vegetation cover is determined based on empirical relations relating the fraction of each plant functional type to mean*

*annual temperature and precipitation. The rooting depth for each plant functional type is a fixed estimate derived from a re-analysis of a global root dataset. Instead in our approach, we do not impose a fixed rooting depth, but it is estimated from the future climate data and our estimate of future transpiration. In the LPJmL4 model (Schaphoff et al., 2018), transpiration depends on the water accessible for plants, which is computed from the relative water content at field capacity and the root distribution within each soil layer. These root distribution estimates are also fixed parameters for each plant functional type considered in the model. Accounting for this climate control on root development and root-zone parameterization in ecohydrological model could potentially also be very interesting (van Oorschot et al., 2021). We will discuss the links with ecohydrological and vegetation models in the revised version of the manuscript.*

**Comment 5:**

l. 58: optimality principles: is this an established term? If not specify what is optimized in this approach(probably it's vegetation growth or something similar)

**Reply 5:**

*We will clarify in the revised version of the manuscript that the optimality principles indeed refer to vegetation growth through optimal allocation of aboveground and belowground resources. This implies that ecosystems have developed root systems to ensure access to sufficient (but not more) water to overcome dry periods (Guswa 2008; Schymanski et al., 2008).*

**Comment 6:**

l. 95: "land-use change under future conditions": The manuscript does not tackle land-use under future conditions. The authors test what happens if land-use is the same in the whole catchment basedon what is already there. But it is never discussed which land-use types are realistic for the future or whether there is a trend in land-use towards any of the present land-use types. Rephrase to make clear that this is just a theoretical assessment of the sensitivity towards different types of land-use instead of a projection into the future. Also, the statement "we exchange space-for-time" (l. 96) suggests, that there is a known land-use trend for the future.

**Reply 6:**

*We agree that we perform a sensitivity assessment of potential/theoretical land-use change and not necessarily projected land-use change and we will rephrase this to "land-use change under potential future conditions". However, the potential changes applied are based on a space-for-time exchange, using characteristics from the Budyko framework of a set of existing catchments to simulate potential changes in a set of different catchments. We believe that the statement "exchange space-for-time" can also be used in case the land-use change for the future is only theoretical.*

**Comment 7:**

The method description is generally a bit confusing. I feel that generally it could be a bit shorter (e.g. the scenario description and the description of the 4 different root-zone storage capacities) are repetitive at some points. It might also help to provide a supportive figure of the study's

workflow that clearly separates between different sources of input data, generation of scenarios and model application (instead of Fig. 3 which would fit better in the Supplemental material). Please revise the method section for more clarity and structure. The specific comments below hopefully help to do that.

*Reply 7:*

*We thank the reviewer for his detailed comments to improve the clarity of the method section. In the revised version of the manuscript, we will try to improve Figure 4 to clarify the workflow, the scenarios and the data used for each scenario. We agree that the current structure is sometimes repetitive, but we think it has the advantage of clearly distinguishing the four different scenarios. Nevertheless, in the revised version of the manuscript, we will try to restructure the Method section in such a way that repetitions are reduced while keeping the distinction in modeling results for each of the scenarios. We agree with the suggestion of the reviewer to move Figure 3 to the Supplement.*

**Comment 8:**

l. 109: "divided into three main zones": It would be nice to see these three main zones in the Figureas well. In the figure, it is unclear which part of the catchment represents which of these three zones.

*Reply 8:*

*Yes, you are right, we will indicate the three zones on the map.*

**Comment 9:**

l. 120: reference is missing for the meteorological variables

*Reply 9:*

*Indeed. The numbers are based on the E-OBS data (Section 3.1) and the historical streamflow data (Section 3.3), we will add these references in the text.*

**Comment 10:**

l. 122: always refer to the specific label of the figure if possible (here it's Fig. 1c and not Fig. 1)

*Reply 10:*

*Agree, we will be more specific.*

**Comment 11:**

l. 147-161: A figure or some numbers comparing the simulated historical and 2K climate scenarios could be a nice addition. From the description, it remains unclear what a "globally 2K warmer world" (l. 158) will translate to in this regional data set. Does this 2K warmer world lead to a mean 2K warmer

regional climate? What's the difference in mean annual temperature and mean annual precipitation in 2K vs. historical climate?

*Reply 11:*

*We agree that a Table summarizing mean annual temperature, potential evaporation and precipitation for the different data sources is a useful addition. Differences in mean annual potential evaporation and precipitation between the simulated 2K and historical climate are now shortly described in the result section 5.1.3 (L396). We will elaborate this further in the revised version of the manuscript.*

**Comment 12:**

l. 164: It would be helpful to add Borgharen to the catchment map in Fig. 1

*Reply 12:*

*Good point, we will add Borgharen on the map of Figure 1.*

**Comment 13:**

Methods: The decision to divide the land-use types into broadleaved forest on the one hand, and coniferous forest/agriculture on the other hand needs better explaining. Why is a tree-dominated (coniferous) vegetation grouped with crops? I would expect that crops and trees are very different with regard to their effect on the water cycle and concerning their root-storage capacity.

*Reply 13:*

*We understand that it may sound confusing. However, the division of both groups was made according to the percentage of broadleaved forest, as we found that omega values tended to be lower for areas with relatively more broadleaved forests (25-38%) in comparison to catchments with relatively low fractions of broadleaved forests (1-12%), as also shown in 5b. We then related this finding to the fact that in the Walloon part of the catchment, most of the old broadleaved forest has been converted to coniferous plantations and agricultural areas, whereas the broadleaved forest mostly remained in the French part of the catchment. In the manuscript, when we refer to "broadleaved forest" versus "coniferous and agriculture", we implicitly mean catchments with relatively high or relatively low percentages of broadleaved forest. However, it is easy to overlook the words "high" and "low" when reading these descriptions, which is why we refer to "broadleaved" and "coniferous and agriculture". We will add a note on this in the revised version of the manuscript.*

**Comment 14:**

l. 233: Why is Imax taken as 2mm?

*Reply 14:*

*We estimate the interception storage capacity (Imax) at 2 mm based on analyses performed in previous studies which report a low sensitivity of the root-zone storage capacity to the value of Imax*

*(de Boer-Euser et al., 2016, Bouaziz et al. 2020). In Bouaziz et al. (2020), we tested the sensitivity of applying interception storage capacities of 0.5, 1.0, 2.0 and 3.0 mm and found a relatively limited impact on the root-zone storage capacity. To reduce the complexity of our analyses, and because of this low sensitivity and our interest in the effect of stationarity versus non stationarity of the root-zone storage capacity in the four scenarios, we decided to use a single value for the interception storage capacity. A single value was also used in van Oorschot et al. (2021). We will include these references in the revised version of the manuscript to explain our choice.*

**Comment 15:**

l. 262: Why are E-OBS data taken from 1980-2018 while streamflow data is only from 2005-2017? Would the results have been different if E-OBS data from 2005-2017 were used instead?

*Reply 15:*

*Thank you for pointing this out. When calculating the root-zone storage capacities, we actually used the period 2005-2017 for both the streamflow data and the meteorological data. This was then not correctly reported in the text, we will make sure to correct this in the revised version of the manuscript. Using the period 2005-2017 or 1980-2017 for the meteorological data in the estimation of $S_{R,max}$ leads to relatively similar ranges of root-zone storage capacities across the scenarios, as shown in Figure 1.*

[Figure]

*Figure 1 Left: Root-zone storage capacities for the 35 catchments of the Meuse basin for the four scenarios derived using meteorological data between 2005-2017 (sane as Figure 5c of the manuscript). Right: Root-zone storage capacities derived using meteorological data between 1980-2017.*

**Comment 16:**

l. 289 ff: How were the ω values sampled?

*Reply 16:*

*When we estimated the root-zone storage capacities for the land-use change scenarios C and D, we estimated the long-term actual evaporation from the Budyko curve through a horizontal shift along the parametric Budyko curve to account for a change in aridity index, and a vertical shift towards a*

*different parametric Budyko curve to account for a change in land-use. For each catchment under change, we assigned an omega value randomly sampled from the set of catchments with current characteristics representing the future characteristics of the catchments under change. We repeated this random sampling seven times, which resulted in seven parameter combinations of $S_{R,max}$ for scenario C and seven parameter combinations for scenario D. We will clarify this in the revised manuscript.*

**Comment 17:**

l. 306: hillslopes are associated with forest and plateau with agriculture. But which type of forest do you mean here? Broadleaved or coniferous?

*Reply 17:*

*The three hydrological response units defined in the hydrological model are determined from topographical data (based on thresholds for Height Above the Nearest Drain and slope) and land-use data (where broadleaved and coniferous forests were both included in the hillslope class, while agricultural land was included in the plateau class). The three classes have slightly different parameterization to reflect different dominant hydrological processes. In the land-use scenarios, we did not change the percentages of each HRU in our model representation. We agree that this is a limitation of our approach, which we will add in the Discussion. However, the data to determine how the link between land-use and HRU may change in the future is not known at this detailed level. Additionally, we expect a limited impact of adapting the fractions of HRU on the hydrological response and we therefore consider this to be an acceptable limitation of our study.*

**Comment 18:**

l. 331ff: "the performance … for the ensemble of retained parameter sets": From the 10000 calibration runs: how many parameter sets were obtained for the model runs? From the supplemental material it looks like the prior is almost the same as the posterior parameter distribution.

*Reply 18:*

*We retained 124 parameter sets based on the defined criteria for model performance. To deal with the relatively long computational costs of running the model, we applied a preliminary first calibration to pre-scan the range of prior distributions. The real calibration was performed with these reduced parameter ranges as prior, which explains the limited difference between prior and posterior distributions.*

**Comment 19:**

l. 334-337: This section can be removed as it is a repetition of what was already mentioned above in lines 272-274.

*Reply 19:*

*We agree and will remove the repetition.*

**Comment 20:**

Scenario description in 4.4:

- It is unclear which values of $S_R$ with regard to the return period are used (2 years or 20 years?)
- How did you decide for the return period in the mixed agricultural/coniferous land-use? Agriculture should be 2 years and forest 20 years (l. 251-253)

*Reply 20:*

*In the distributed model, each cell has a percentage wetland, hillslope and plateau. The root-zone storage capacity parameter for the wetland and plateau hydrological response units were assigned a return period of 2 years, while a return period of 20 years was assigned to hillslope. We refer to the studies of Nijzink et al. (2016) and Gao et al. (2014) where return periods of 20 years are associated with forested areas. Lower return periods of 2 years are better suited for agricultural areas (Wang-Erlandsson et al. 2016). We will clarify this in the revised version of the manuscript.*

**Comment 21:**

l. 357, 362, 369: no need to repeat that $S_{Rmax,a}$ is used as a parameter in the historical run for every scenario. Better to mention it once, when the historical run is explained.

*Reply 21:*

*Agree, we will adapt this in the revised version of the manuscript.*

**Comment 22:**

Results: It is not always clear what the reported numbers represent. Median and standard deviation? Mean and standard error of the mean? E.g. l. 374 & 377, l. 382, l. 390 & 391, l. 402, l. 408.If the reported values are always the same, you could also mention it once and state that all subsequent values represent the same measures.

*Reply 22:*

*Good point, the reported numbers represent the median and standard deviation, we will make sure to mention this once clearly.*

**Comment 23:**

l. 376: should this be $\omega_{obs}$ instead of $\omega$?

*Reply 23:*

*Correct, we will adapt this.*

**Comment 24:**

l. 377: should this be transpiration instead of evaporation? This is a general issue: there is no clear distinction between evaporation, transpiration and evapotranspiration in the text.

*Reply 24:*

*Throughout the manuscript, we use the term evaporation to represent all the different evaporation components (interception, transpiration and soil evaporation). It is perhaps a matter of taste, but we like to follow the terminology proposed by Savenije (2004) and Miralles et al. (2020), where evaporation instead of evapotranspiration is used to refer to all evaporative fluxes.*

**Comment 25:**

l. 377-379: The differences of $\omega$ between the catchments is mainly attributed to the differences in the main vegetation type (broadleaved vs. coniferous/agriculture, l. 377-379). However, the catchments also differ substantially in other characteristics (French part: thick soils and gentle slopes, thin soils and steep terrain in the Ardennes, porous chalk in Wallonia (l. 109-113)). It should be discussed to what extent the differences in $\omega$ might not be dependent on the vegetation cover alone but also on the topography and soil type/thickness. Also, what are the implications of this regarding the method?How sure are you that the differences in hydrology between land-use types are really caused by the vegetation cover and not by the underlying topographical and soil characteristics?

*Reply 25:*

*This is an interesting question which we will include in the discussion of the revised version of the manuscript. The differences in omega-values are most probably related to a combination of biophysical features. However, considering that transpiration is the largest continental water flux (Jasechko, 2018) and that omega values determine the hydrological partitioning, we assume that the variability in omega values is largely controlled by the root-accessible water volume $S_{R,max}$. This root-accessible water volume is independent from the soil type, as root systems will develop in a way to ensure sufficient access to water. In clayey soils, the rooting depth might be shallower than in sandy soils for an identical root-zone storage capacity. In our opinion, geology, soils characteristics and topography are implicitly integrated in other model parameters, e.g. the time scales of the linear reservoirs which represent the subsurface flow resistance in different parts of the system.*

**Comment 26:**

l. 394: Fig. 2b should either be referenced earlier in the text, e.g. when talking about the difference between the historical and the 2K climate time series in the method section or it should be a separateresult figure that comes later in the text.

*Reply 26:*

*Is it perhaps possible that you overlooked the reference to Fig 2b earlier in the text in Section 4.1.2 (L243) to illustrate the water-balance approach to estimate the root-zone storage capacity?*

**Comment 27:**

l. 424: "median values of approximately 0.93": why approximately?

*Reply 27:*

*You are right, we will remove 'approximately'*

**Comment 28:**

l. 431: "streamflow during the wettest months": include which months you mean by "wettest months"

*Reply 28:*

*Good point, we will clarify that here we refer to the months December and January as wettest months.*

**Comment 29:**

l. 500: "shows distinct patterns of change": more precise language could be used: Which response variables differ and are they larger or smaller compared to the stationary scenario?

*Reply 29:*

*This is a good suggestion, we had not included more details to avoid repetition from the result section. However, we think changes in streamflow and evaporation can briefly be repeated here to be more precise. We will clarify this in the revised manuscript.*

**Comment 30:**

l. 512-519: This section does not fit in the "Implications" section of the discussion. It is more of a limitation of the current study or an outlook of what could be done next. It could e.g. be moved to the"Limitations and Knowledge gaps" section of the discussion.

*Reply 30:*

*We agree that this paragraph contains an outlook of what could be done next. Initially, we had seen this as an implication of our work for future work, but we agree that it would better fit in a "Limitations and outlook" section of the discussion. We will adapt this in the revised version.*

**Comment 31:**

l. 524-256: It is not clear to me, why the results on actual evaporation differences between the scenarios indicate disagreements among model process representations. Please elaborate more on this point. Also, what are the specific "processes that become relevant in the future"?

*Reply 31:*

*What we mean here is that in the future scenario, evaporation demand increases. In scenario $2K_A$, where the root-zone storage capacity has not adapted to the future climate, we see water stress conditions that do not occur in the other scenarios. The different model representations amongst scenarios lead to different hydrological responses. However, we might consider removing this point in the revised version of the manuscript and add the other relevant points of discussion mentioned earlier in our reply to Comment 2.*

**Comment 32:**

l. 333-334: The conclusion, that vegetation is important for regulating the water cycle is correct but itis also quite established and not really a specific discussion of your results.

*Reply 32:*

*We agree that this conclusion is already quite established. We will rephrase this statement to emphasize how our study contributes to the quantification of the potential impact of vegetation adaptation in regulating the water cycle.*

**Comment 33:**

l. 535-539: This discussion is also a limitation of your study or an outlook to further work. It should notbe under the "Implications" subheading of your discussion.

*Reply 33:*

*We agree that this part of the discussion is also more an outlook for future research and will move this to the Limitation and outlook section.*

**Comment 34:**

l. 542: "it is unclear how ecosystems will cope with climate change": A discussion of how useful your approach to include vegetation into hydrological models under climate change in the light of this uncertainty would be interesting. To what extent can we be sure that the root-zone storage capacity can adapt to changing climate? What evidence is there from other studies regarding this issue? How would you proceed with your approach if vegetation changes to a vegetation type for which there is no data from the same region?

*Reply 34:*

*This is a very interesting point. There is increasing evidence that vegetation efficiently adapts its root-zone storage capacity to ensure sufficient access to water (Guswa 2008, Schymanski et al. 2008). However, while we know that the ecosystem will eventually adapt to changing environmental conditions, partly by changing the mix of vegetation species and partly by vegetation adjusting its rooting depth or density, the question is how long it will take for an ecosystem to adapt in relation to the rate of climate change. Also, there are limits to the capacity of an ecosystem to adapt, for instance when is the threshold passed for the adaptability of rainforest to become savannah, or where lies the threshold for savannah to become desert? In this study we assume that adaptation thresholds are not reached. We refer to our reply to comment 2 for further details on this matter.*

*An interesting next step for our methodology will be to apply it in a climate-matching approach (Fitzpatrick and Dunne, 2019), where the current climate and landscape characteristics of a location X match the future climate or landscape characteristics of a location Y. This climate matching could be applied over distant regions, using datasets which combine landscape and climatological data over large samples of catchments (e.g. the various CAMELS datasets). Despite considerable uncertainties, this may allow us to infer vegetation adaptation and the associated changes in root-zone storage capacity from identifying regions in the world where the current climate resembles the projected future climate in a different region.*

**Comment 35:**

At the end of the discussion, you mention several times that this study should be read as a sensitivityanalysis (l. 571) and a proof-of-concept (l. 575). This should also be made clear in the abstract. Also, athorough discussion of the advantages and disadvantages of the presented method is missing. What are possible applications of it, to what types of regions/questions can it be applied? What are the limitation and what could be improved?

*Reply 35:*

*This is a very good suggestion, in the revised abstract, we will more strongly emphasize that our study should be understood as a sensitivity analysis. As also mentioned in our replies to the main comments (1 and 2), we will not use the terminology "proof-of-concept" anymore as we cannot test our results against future observations. We agree with your suggestion to more thoroughly discuss the advantages and disadvantages of the presented method in the discussion. We refer to our detailed reply to Comment 2 for the specific points that we will address.*

**Comment 36:**

Figure labels should be in the same position for all figures (e.g. top left)

*Reply 36:*

*Agree, we will adapt this in the revised version.*

**Comment 37:**

Figure labels could be bold for better visibility?

*Reply 37:*

*Good suggestion, we will adapt this in the revised version.*

**Comment 38:**

Why are the scenario names (2Ka-d) that are defined in Fig. 4 never used? Instead $S_{rmaxa-d}$ is used in Figs. 5,8,9? If scenario names are given, they should be used consistently.

*Reply 38:*

*Very good point. In Figure 9, we are actually showing values of $S_{R,max}$. However, in Figure 8 and 9, it indeed makes more sense to refer to Scenario 2KA etc in the labels.*

**Comment 39:**

Fig. 1:
- colours of figure b): better use some continuous colour scheme
- Figure labels are inconsistent, b and c not on the same height
- Fig. 1b: what are the black points? Are they the streamflow measurement locations? Mention in the caption
- Fig. 1 does not reflect well many aspects mentioned in the text (2.1 landscape and 2.2 landuse)
    - Which are the three zones mentioned in l. 109? Are they represented in Fig. 1b? If yes you could add this to the caption. It is not clear what is the French, the Ardennes and the Wallonia part mentioned several times in the text
    - Fig. 1b: The numbers don't really match with the text. In Walloon 44% of the broadleaved forest should be there (l. 126), but in the figure the max. percentage is 38%.

*Reply 39:*

- *We will test if an alternative color scheme improves readability.*
- *We will move the labels*
- *The black points are indeed the streamflow measurement locations, we will add this in the*

*caption.*
- *We will add the location of the three zones*
- *When we refer to 44% in the text, we mean 44% of the 18th century Walloon forests of Belgium that have remained from the original broadleaved forests. The 38% in the figure refer to the fraction of broadleaved forest within a catchment.*

**Comment 40:**

Fig. 3:
- Maybe this figure fits better in Supplement S3 because it is part of the model description? Idon't find it very helpful in the manuscript without the context of the model formulas

**Reply 40:**

*We agree that Figure 3 can be moved to the Supplement to be connected to the model description. We will modify this in the adapted version.*

**Comment 41:**

Fig. 5:
- Labels are missing
- Figures are a bit small: Could be a made bigger if empty space between panels is reduced
- 5b:
  - $\omega_{obs}$ should be on the y-axis not just $\omega$
  - Axis text: No % because it's already in x-axis title
- 5c:
  - Caption last sentence: "A similar but reversed approach is applied …" It is the *same* and not a *similar* approach that was used.

**Reply 41:**

- *Indeed, we will add the labels in the revised version.*
- *We will try to decrease the empty space between the panels to increase the panels themselves.*
- *We will replace $\omega$ by $\omega_{obs}$*
- *We will remove % from the x-axis title*
- *Indeed, we will replace similar by same.*

**Comment 42:**

Fig. 6:
- What is the ribbon for the modelled values: range from all realistic parameter sets of thecalibration?

**Reply 42:**

*Indeed, the ribbon represents the ensemble of feasible parameter sets, we will clarify this in the caption.*

**Comment 43:**

Fig. 7:
- Could be larger: box is not visible
- Don't use transparent colours to distinguish the panels. In my opinion they are alreadydistinguished enough by the panel titles and labels in the caption (same for figures in Supplement S3)
- Labelling is not consistent (compare to labelling of Fig. 6)
- Why is there such a big difference between Borgharen and the 34 catchments? Isn'tBorgharen just a summary of all the catchments?

*Reply 43:*

- *It is more the shape of the violin plots (left and right) which are important here.*
- *We consistently applied a color code throughout the Figures and would like to keep it as we believe it increases the clarity.*
- *We will change the labeling order.*
- *Borgharen is the most downstream outlet point considered. Often, model performance tends to decrease for smaller catchments. Additionally, the calibration was performed at Borgharen.*

**Comment 44:**

Fig. 8:
- Caption 8e) maybe mention that y-axis is different scale (compare to caption of Fig. 7)

*Reply 44:*

*Yes, we will add this in the revised version.*

**Comment 45:**

Fig. 9:
- What are the ribbons and lines? Median + conf. interval?

*Reply 45:*

*Good point, they indeed show median and range of ensemble retained sets, we will clarify this in the caption.*

**Comment 46:**

S1: Monthly correction factors for E-OBS precipitation data
- First sentence: Citation missing

*Reply 46:*

*Indeed, we will add the missing reference.*

**Comment 47:**

S4: Prior and posterior parameter distributions
- State in table heading, that the last 3 columns are the posterior parameter distributions

*Reply 47:*

*Yes, we will add this in the revised version.*

**Comment 48:**

l. 54: rephrase to: sensitivity of the hydrological response to change in …

*Reply 48:*

*Yes, we will rephrase.*

**Comment 49:**

l. 62: remove "as often referred to"

*Reply 49:*

*Agree.*

**Comment 50:**

l. 79: remove the full stop before the list of references

*Reply 50:*

*Yes.*

**Comment 51:**

l. 191 & 1197: same style for (p1), (p2) and p3 (either with or without brackets)

*Reply 51:*

*Yes, we will make this consistent in the revised version.*

**Comment 52:**

l. 392: replace "return periods of 2 year" with either "2 year return period" or "return period of

2years". Also check the subsequent text as this mistake happens several times.

*Reply 52:*

*Good point, we will replace.*

**Comment 53:**

l. 410: Vertical space is missing as a new paragraph begins in line 411

*Reply 53:*

*Not sure what is meant here, the spacing looks the same as in the other paragraphs.*

**Comment 54:**

l. 500: "compared to" instead of "with respect to"?

*Reply 54:*

*Ok, we will adapt.*

**Comment 55:**

l. 592: "distinct change of sign": remove distinct

*Reply 55:*

*Agreed.*

**Comment 56:**

Avoid unspecific adverbs. Either remove them, or state specifically what you mean by them. E.g.
- l.114: "relatively short response time" (how short is relatively short?)
- l. 422: "relatively well reproduced"
- l.423: "slight underestimation" and "relatively similar performance"

*Reply 56:*

*L114, we will be more specific about the response time in the revised version.*
*L422 and 423, numbers are given later in the sentence, we will clarify this in the revised version.*

**References**

Berghuijs, W. R., Gnann, S. J., & Woods, R. A. (2020). Unanswered questions on the Budyko framework. *Hydrological Processes*, (October), 1–5. https://doi.org/10.1002/hyp.13958

de Boer-Euser, T., McMillan, H. K., Hrachowitz, M., Winsemius, H. C., & Savenije, H. H. G. (2016). Influence of soil and climate on root zone storage capacity. *Water Resources Research*. https://doi.org/10.1002/2015WR018115

Bouaziz, L. J. E., Steele-Dunne, S. C., Schellekens, J., Weerts, A. H., Stam, J., Sprokkereef, E., et al. (2020). Improved understanding of the link between catchment-scale vegetation accessible storage and satellite-derived Soil Water Index. *Water Resources Research*. https://doi.org/10.1029/2019WR026365

Brunner, I., Herzog, C., Dawes, M. A., Arend, M., & Sperisen, C. (2015). How tree roots respond to drought. *Frontiers in Plant Science*, *6*(JULY), 1–16. https://doi.org/10.3389/fpls.2015.00547

Donohue, R. J., Roderick, M. L., & McVicar, T. R. (2012). Roots, storms and soil pores: Incorporating key ecohydrological processes into Budyko's hydrological model. *Journal of Hydrology*, *436–437*, 35–50. https://doi.org/10.1016/j.jhydrol.2012.02.033

Donohue, R. J., Roderick, M. L., McVicar, T. R., & Farquhar, G. D. (2013). Impact of CO2 fertilization on maximum foliage cover across the globe's warm, arid environments. *Geophysical Research Letters*, *40*(12), 3031–3035. https://doi.org/10.1002/grl.50563

Fan, Y., Miguez-Macho, G., Jobbágy, E. G., Jackson, R. B., & Otero-Casal, C. (2017). Hydrologic regulation of plant rooting depth. *Proceedings of the National Academy of Sciences*, 201712381. https://doi.org/10.1073/pnas.1712381114

Fitzpatrick, M. C., & Dunn, R. R. (2019). Contemporary climatic analogs for 540 North American urban areas in the late 21st century. *Nature Communications*, *10*(1), 1–7. https://doi.org/10.1038/s41467-019-08540-3

Frank, D. C., Poulter, B., Saurer, M., Esper, J., Huntingford, C., Helle, G., et al. (2015). Water-use efficiency and transpiration across European forests during the Anthropocene. *Nature Climate Change*, *5*(6), 579–583. https://doi.org/10.1038/nclimate2614

Gao, H., Hrachowitz, M., Schymanski, S. J., Fenicia, F., Sriwongsitanon, N., & Savenije, H. H. G. (2014). Climate controls how ecosystems size the root zone storage capacity at catchment scale. *Geophysical Research Letters*, *41*(22), 7916–7923. https://doi.org/10.1002/2014GL061668

Gentine, P., D'Odorico, P., Lintner, B. R., Sivandran, G., & Salvucci, G. (2012). Interdependence of climate, soil, and vegetation as constrained by the Budyko curve. *Geophysical Research Letters*, *39*(19), 2–7. https://doi.org/10.1029/2012GL053492

Guswa, A. J. (2008). The influence of climate on root depth: A carbon cost-benefit analysis. *Water Resources Research*, *44*(2), 1–11. https://doi.org/10.1029/2007WR006384

Hrachowitz, M., Stockinger, M., Coenders-Gerrits, M., van der Ent, R., Bogena, H., Lücke, A., & Stumpp, C. (2020). Deforestation reduces the vegetation-accessible water storage in the unsaturated soil and affects catchment travel time distributions and young water fractions. *Hydrology and Earth System Sciences*, *i*(June), 1–43. https://doi.org/10.5194/hess-2020-293

Jasechko, S. (2018). Plants turn on the tap. Nature Climate Change, 8, 560–563.

Miralles, D. G., Brutsaert, W., Dolman, A. J., & Gash, J. H. (2020). On the use of the term "Evapotranspiration." *Earth and Space Science Open Archive*, 8. https://doi.org/10.1002/essoar.10503229.1

Nijzink, R., Hutton, C., Pechlivanidis, I., Capell, R., Arheimer, B., Freer, J., et al. (2016). The evolution of root-zone moisture capacities after deforestation: A step towards hydrological predictions under change? *Hydrology and Earth System Sciences*, *20*(12), 4775–4799. https://doi.org/10.5194/hess-20-4775-2016

van Oorschot, F., van der Ent, R., Hrachowitz, M., & Alessandri, A. (2021). Climate controlled root zone parameters show potential to improve water flux simulations by land surface models. *Earth System Dynamics Discussions*, 1–26. https://doi.org/10.5194/esd-2021-3

Reaver, N., Kaplan, D., Klammler, H., & Jawitz, J. (2020). Reinterpreting the Budyko Framework. *Hydrology and Earth System Sciences Discussions*, (November), 1–31. https://doi.org/10.5194/hess-2020-584

Savenije, H. H. G. (2004). The importance of interception and why we should delete the term evapotranspiration from our vocabulary. *Hydrological Processes*, *18*(8), 1507–1511. https://doi.org/10.1002/hyp.5563

Schaphoff, S., Von Bloh, W., Rammig, A., Thonicke, K., Biemans, H., Forkel, M., et al. (2018). LPJmL4 - A dynamic global vegetation model with managed land - Part 1: Model description. *Geoscientific Model Development*, *11*(4), 1343–1375. https://doi.org/10.5194/gmd-11-1343-2018

Schymanski, S. J., Sivapalan, M., Roderick, M. L., Beringer, J., & Hutley, L. B. (2008). An optimality-based model of the coupled soil moisture and root dynamics. *Hydrology and Earth System Sciences*, *12*(3), 913–932. https://doi.org/10.5194/hess-12-913-2008

Singh, C., Wang-Erlandsson, L., Fetzer, I., Rockström, J., & van der Ent, R. (2020). Rootzone storage capacity reveals drought coping strategies along rainforest-savanna transitions. *Environmental Research Letters*. https://doi.org/10.1088/1748-9326/abc377

Tietjen, B., Schlaepfer, D. R., Bradford, J. B., Lauenroth, W. K., Hall, S. A., Duniway, M. C., et al. (2017). Climate change-induced vegetation shifts lead to more ecological droughts despite projected rainfall increases in many global temperate drylands. *Global Change Biology*, *23*(7), 2743–2754. https://doi.org/10.1111/gcb.13598

Troch, P. A., Carrillo, G., Sivapalan, M., Wagener, T., & Sawicz, K. (2013). Climate-vegetation-soil interactions and long-term hydrologic partitioning: Signatures of catchment co-evolution. *Hydrology and Earth System Sciences*, *17*(6), 2209–2217. https://doi.org/10.5194/hess-17-2209-2013

Wang-Erlandsson, L., Bastiaanssen, W. G. M., Gao, H., Jägermeyr, J., Senay, G. B., Van Dijk, A. I. J. M., et al. (2016). Global root zone storage capacity from satellite-based evaporation. *Hydrology and Earth System Sciences*, *20*(4), 1459–1481. https://doi.org/10.5194/hess-20-1459-2016

Yang, Y., Roderick, M. L., Zhang, S., McVicar, T. R., & Donohue, R. J. (2019). Hydrologic implications of vegetation response to elevated CO2 in climate projections. *Nature Climate Change*, *9*(1), 44–48. https://doi.org/10.1038/s41558-018-0361-0

Zhang, B., Hautier, Y., Tan, X., You, C., Cadotte, M. W., Chu, C., et al. (2020). Species responses to changing precipitation depend on trait plasticity rather than trait means and intraspecific variation. *Functional Ecology*, (September), 2622–2633. https://doi.org/10.1111/1365-2435.13675

---

## Author Comment (AC2)

**Comment:**

This manuscript evaluates how predicted changes in climate, e.g. aridity and seasonality, reflect on catchment hydrology in an exemplary basin using a hydrological model. The novelty lies in the accounting for the necessary adaptation in the vegetation root zone storage (essentially rooting depth) to actually satisfy predicted changes in actual evapotranspiration. For this, the authors first establish the expected rooting depth required to satisfy evapotranspiration due to climatic shifts of precipitation, evapotranspiration and their timing. Next they use those in a hydrological model to show that vegetation root adaptation and to a lesser extent als land use changes have a discernible effect on predicted catchment water balance. The authors conclude that this study serves as a proof of concept that adaptive vegetation has to be considered when evaluating climate change effects on hydrology. I agree with this conclusion and believe (although I have some questions) that the methodology is suitable to make this statement. I think this is a valuable contribution and of interest to the readership of HESS.The manuscript is formulated grammatically well. Having said this, it does not read well, for reasons stated below and requires revision. I fact, I really had to fight my way throughthe methods section. I alos have some serious concerns on lack of information and general organization of the manuscript. I recommend major revisions.

*Reply:*

*We appreciate the reviewer's overall positive assessment of the manuscript and we are thankful for his/her thoughtful comments. We provide detailed clarifications below on how we will revise the manuscript.*

**Comment:**

I have some concerns about missing information or implications of some assumptions that prevents me from fully evaluating the results.

I find it difficult to understand how the evapotranspiration was estimated for the model,and this needs to be laid out more clearly. For the rooting depth estimation ET from theroot zone was derived from applying the observed \omega to the predicted potential ET. But what was used for forcing the hydrological model? Potential ET from the climateprediction? What happens in the hydrological model, when the root zone storage runs dry? I read in the discussion that water limitation reflects on ET, but there is no mention how?

*Reply*

*The three inputs used to force the hydrological model are indeed potential evaporation, temperature and precipitation from the observed and simulated historical and future climate data. We will make sure to clearly state this in the model description of the revised version of the manuscript. For the actual evaporation from the root-zone storage in the hydrological model, we apply a simple formulation to express water stress. The equation is provided in Table S4 of the Supplementary*

*material and describes how actual evaporation is linearly reduced when the root-zone storage is below a certain threshold (parameter). This standard formulation is used in many conceptual models, including HBV, NAM and VHM (Bouaziz et al., 2021). We will clarify this in the revised version of the manuscript.*

**Comment:**

Fig 3 is very repetitive, while the essential difference between the hillslope, plateau andwetlands is difficult to spot: It is whether or not the model allows for ground water exchange. Now, since the vegetation types are attributed to either hillslope (broadleaved forest) and plateau (conifers, agriculture) this small detail becomes important (and should be spelled out). How is this accounted for when the vegetation isswapped? Are also the HRUs swapped, e.g. does the area capable of ground water recharge increase / decrease as a result of the swap? In other words, does the model structure change as a result of the swap?

**Reply**

*This is an interesting suggestion. However, in the land-use change scenarios, we did not change the percentages of each HRU in our model structure. The approach we propose to estimate the effect of land-use change really is a top-down approach based on assumed trajectories within the Budyko framework, but without the level of detailed required to specifically change land-use type at the pixel level. Therefore, we did not have the data available to change the percentage of each Hydrological Response Unit in our theoretical land-use change experiments. However, we think it is a good suggestion to discuss this limitation in the discussion section of the revised manuscript. We will also make sure to add a more detailed model description in the Supplement of the revised version of the manuscript, and to clarify the main differences in the caption of the Figure. In relation with the comments from Referee #1, we propose to move Figure 3 to the Supplement.*

**Comment:**

As a follow up on that, I was left unclear as to whether all 2K scenarios see the same climate forcing? Does the change in \omega only apply to the rooting depth parameteror also to the evapotranspiration forcing? Please spell this out.

**Reply**

*In the revised version of the manuscript, we will clarify that the same climate forcing is used for each of the 2K scenarios. The change in omega in combination with the change in climate data are indeed translated to a change in root-zone storage capacity parameter. However, we did not change the potential evaporation. We will clarify this in the revised version of the manuscript.*

**Comment:**

I would appreciate an extension of the discussion to critically review the results.

a) The discussion already has a section called „limitations", which is good. But it should include some more discussion on the assumptions above.

b) Correlations between parameters / vegetation and the environment are neglected in this study. For example, could the differences in \omega between catchments in Franceand the Belgium partly be related to differences in geology, topography etc. besides forest cover? Can you safely assume that the calibrated catchment parameters obtained for a specific vegetation distribution are still valid when changing the vegetation? I agree with the general statement that this a modeling study to provide a proof of concept, but would be good to include this in the discussion.

c) The manuscript starts with hypotheses which is nice and suitable for this study. It would be good to come back to them specifically in an interpretation section of the discussion.

*Reply*

a) *We agree with the suggestions of the Referee #1 and Referee #2 to revise the Discussion Section in the revised version of the manuscript.*

b) *This is in an interesting question, which we will discuss in the revised version of the manuscript. We agree that the differences in omega-values are most likely related to a combination of biophysical characteristics. However, the omega parameter describes the hydrological partitioning and because transpiration is the largest continental flux, we think it is reasonable to assume that land use plays a major role to explain the differences in omega values (Teuling et al., 2019). Therefore, the variability in omega values is largely controlled by the water volume accessible to the roots of vegetation for transpiration (i.e. $S_{R,max}$). Topography, geology and soil type are likely implicitly integrated in other model parameters, e.g. the various recession time-scales of the linear reservoirs, which represent subsurface flow resistance throughout the system. We will include this in the revised discussion.*

c) *Thank you, this is a good point and we will make sure to clearly come back to the hypothesis in the Discussion of the revised manuscript.*

**Comment:**

The manuscript reads technical at many levels, and this seriously prevents communicationto the point where important information seems to be missing. For example,

a) The introduction of the simulated climate in section 3.2. gives information about the origin of the time series, but leaves out which variables were actually used in the study. Specifically, the reader is left to guess whether it is potential ET or actual ET ?

b) Similarly, the structure of the hydrological model is shown in Fig 3, and given in a very short section 4.2. The model description does not include a reference to how root zone storage affects actual evapotranspiration. In this study on rooting depth and effects onthe water cycle this is a central point and should not be left out. It is only mentioned (Ibelieve once) in the discussion.

c) I am assuming that two parameters for the root zone storage capacity are used in each model run, one for shallow (agriculture and coniferous forest) and one for deeper rooted (broadleaf forests) vegetation. I am not sure whether I overlooked this, but it would be good to spell this out in the section where the model or the calibration are introduced.

*Reply*

a) *In the revised manuscript, we will clarify that potential evaporation is used to force the model and that actual evaporation is estimated in the model, following the equations provided in the Supplement.*

b) *The equations that describe how root-zone storage affects transpiration are included in the supplement. Following the suggestion of Referee #1, we will move Fig 3 to the Supplement of the revised version and include a more detailed description of the modelled processes.*

c) *For the root-zone storage capacity parameter, we use a return period of 2 years for the wetland and plateau classes and a return period of 20 years for the hillslope HRU. This is indeed already mentioned in the Model calibration section (4.3.1).*

**Comment:**

There are plenty of abbreviations that are barely introduced, sometimes the introduction appears even in a subheading.

*Reply*

*Thank you for pointing this out, we will make sure to clarify the abbreviations in the revised version of the manuscript.*

**Comment:**

The order within the methods section prevents understanding the methods. For example, there are many references to the model runs, before the model structure iseven introduced. Therefore it is really difficult to digest the information or interpret what the assumptions mean for the model. etc.

*Reply*

*In the revised version of the manuscript, we will try to restructure the Method section to clarify our approach and experiments. We think that improving Figure 4 and introducing it earlier in the manuscript (perhaps already at the start of the Method section) can potentially also improve the clarity of the reading.*

**Comment:**

Currently the headings and subheadings are not suitable for a reader navigating the text. Consider that they should help finding information when the reader does not diveinto the main text completely. For example take section „4.1.2 Seasonal water balancefor estimating the change in root zone storage capacity S_R,max", would be more easily called „4.1.2 Estimation of root zone storage capacity". I could make such propositions for almost every heading. Please revise.

*Reply*

*We thank the reviewer for this good suggestion, we will go through the section titles and simplify them in the revised version of the manuscript.*

**Comment:**

It is difficult to interpret the results without a table showing an overview of the climateof the different scenarios, e.g. precipitation, E_pot, aridity, seasonality, if applicable actual evapotranspiration used as forcing, actual evapotranspiration as model output.

*Reply*

*We agree with this suggestion of the referee and will include such a table in the revised version of the manuscript.*

**Comment:**

I believe the manuscript can be shortened and the important information be fleshed outto improve it being understood.

*Reply*

*This is a good point and we will critically go through our manuscript to see which part of the analyses can possibly be moved to the Supplement to cut down on some technical details. We will also revise the Discussion to include additional implications and limitations as suggested by the referees.*

**Comment:**

L 8-10: Needs to become obvious that these are modeling hypotheses. Please reformulate

*Reply*

*We will clarify that our hypothesis relates to a modeling study.*

**Comment:**

L 14-15: At this point in the manuscript it is difficult to understand why those particular changes are considered. Maybe formulate more general

*Reply*

*We agree and will remove "from coniferous plantations/agriculture towards broadleaved forest and vice versa" and only keep the "two hypothetical changes in land use.".*

**Comment:**

L 17-18 Are these numbers consistent with the water balance? They do not look like they do ..

*Reply*

*The numbers mentioned here reflect the mean differences between, on the one hand, the change in mean annual streamflow and evaporation between the scenarios $2K_B$, $2K_C$, $2K_D$ with adaptive ecosystems and, on the other hand, the stationary scenario $2K_A$ (see section 5.3.5). Therefore, these fractions do not relate to the total water balance. We will rephrase these sentences: "We found that the larger root-zone storage capacities (+34%) in response to a more pronounced seasonality with drier summers under 2K global warming strongly alter seasonal patterns of the hydrological response. The differences in the change of mean annual evaporation, recharge and streamflow between, on the one hand, the three scenarios with adaptive root-zone storage capacity and, on the other hand, the stationary system are +4%, -6% and -7%, respectively."*

**Comment:**

L 25-27: There should be more appropriate references for this very general comment.

*Reply*

*We will add additional references on the increasing evidence that ecosystems have the capacity to adapt to local (and changing) climate conditions, including Guswa, 2008; Schymanski et al., 2008; Gentine et al., 2012; Harman and Troch, 2014; Hrachowitz et al., 2020.*

**Comment:**

L 42: „stationarity is dead" - use citation marks, otherwise it seems a bit awkward language, as strictly speaking stationarity never lived.

*Reply*

*We agree and will add the citation marks.*

**Comment:**

L 55: „require …" I do not agree. In a distributed model it could also s just be represented by distribution of land cover. This does not require a priori knowledge of the relation to catchment outflow.

*Reply*

*We are not sure to understand the comment made by the reviewer, but what we mean is that in a*

*distributed model, the land cover map somehow needs to be translated to parameter values. Often, look-up tables retrieved from literature are used to relate a specific land use to a model parameter value. An alternative approach is to transfer parameters values from one location to another location through regionalization approaches. However, there is considerable uncertainty in both of these a priori parameter estimations.*

**Comment:**

L 56 „uncertainty in ..“ this statement is very vague. Can you be more explicit?

*Reply*

*See our reply to the previous comment. We will try to be more explicit in the revised version of the manuscript.*

**Comment:**

L 72-25: As it stands, this appears quite unrelated. Either erase or put into context.

*Reply*

*We agree and we will rephrase to clarify the context in the revised manuscript to better introduce this paragraph. Here, we want to emphasize that there are multiple factors, besides the aridity index, affecting the position of a catchment in the Budyko space. One of these factors relates to the responses of ecosystems to elevated $CO_2$ levels, which are complex and can counteract one another (Jasechko 2018). On the one hand, vegetation density may increase from $CO_2$ fertilization, leading to increased transpiration. On the other hand, higher water use efficiencies may lead to declining transpiration rates as plants may transpire less water per unit of $CO_2$ taken up.*

**Comment:**

L 76: „match expectations of the Budyko curve“ - Unclear, please be more specific: Which expectations?

*Reply*

*We agree and we will rephrase. What we mean here is that the fact that most catchments worldwide scatter closely around the analytical Budyko curve is evidence for the co-evolution of catchment vegetation and soils with climate.*

**Comment:**

L 77-78: „Vegetation tends to efficiently adapt its root-zone storage capacity to satisfycanopy water demand." - reference needed, ideally with an observation component.

*Reply*

*Good point, actually the references are mentioned after the next sentence, but we will move them earlier in the revised version of the manuscript.*

**Comment:**

L 78-79: I believe Yang et al., 2016 wold be good to cite here

*Reply*

*We thank the referee for this very interesting reference, which we will include.*

**Comment:**

L 95-98: Very difficult to grasp. I am not sure whether this paragraph really helps to understand what is coming.

*Reply*

*We will rephrase the last two paragraphs of the introduction to emphasize that current studies assume that model parameter remain constant in a changing system. The objective of our study is to test how sensitive hydrological predictions are when changing vegetation related parameters, thereby accounting for the adaption of vegetation to future climate conditions.*

**Comment:**

L 99-100: Any reasons for this hypothesis? Also, would be good to come back to it specifically in the discussion.

*Reply*

*This is a good suggestion, we will clarify in the discussion that we expect the changes in the predicted hydrological response as a result of 2K global warming to be more pronounced in comparison to current-day conditions due to the potentially drier and warmer summers.*

**Comment:**

102-105: Again, not sure this really helps. It is too detailed to soon.

*Reply*

*Agree, we will remove these lines in the revised version of the manuscript.*

**Comment:**

L 119-120: Reference missing

*Reply*

*You are right, in the revised version, we will clarify that these numbers are calculated from the observed historical E-OBS data (section 3.1) and the streamflow data at Borgharen (section 3.3).*

**Comment:**

L 131: Reference on the biodiversity statement required.

*Reply*

*Agreed, we will add the reference of Kervyn et al. (2018) here.*

**Comment:**

L 138: „E-Obs" Add definition also in the text, not only in subtitle

*Reply*

*Thanks, we will clarify this.*

**Comment:**

L 147: Same as above, please introduce abbreviations in the text before using them. Also,with 2 K you probably refer to 2 Kelvin. Please spell this out as well.

*Reply*

*Agreed.*

**Comment:**

L 150: Spell out RACMO2 and HTESSEL ?

*Reply*

*Agreed.*

**Comment:**

L 168-180: Generally, it is a good idea to explain what is coming, but I really did not getit.
Maybe try rewording in plainer language and less specific?

*Reply*

*We will try to rephrase this paragraph to increase readability. In the first sentence, we will explicitly refer to the change in vegetation related parameters in hydrological models (more specifically the root-zone storage capacity) in response to climate change. We will try to be less specific to clarify the broader picture.*

**Comment:**

L 193-195: Here I was entirely confused. Is \omega_{change} derived from the climate data using E_A from there? This part is very opaque, but really critical to understanding the methods.

*Reply*

*We will rephrase to clarify that the long-term $E_A$ is derived from trajectories in the Budyko space considering a change in aridity index (from the climate data) and a potential change in omega-values. The change in omega-values are derived from historical omega-values in catchments with relatively high and relatively low percentages of broadleaved forests.*

**Comment:**

L203-206: Can you be sure that the runoff coefficients only depend on the forest cover and not on the geology? It seems that the regions with high / low cover are geographicallydistinct. How to avoid misinterpretation?

*Reply*

*In the revised version of the manuscript, we will acknowledge that runoff coefficients are of course also related to other physical catchment characteristics besides land cover. However, as transpiration is the largest continental flux, we assume that vegetation plays a major role in the hydrological partitioning. There is increasing evidence that vegetation develops root systems in an optimal way to fulfill their needs. It is important to make the distinction between rooting depth and root-zone storage capacity. The root-zone storage capacity is independent from the soil type, as in clayey soils the rooting depth may be shallower than in sandy soils for an identical root-zone storage capacity (= root-accessible water volume). Geology also plays an important role in the hydrological response but*

*is likely implicit in other model parameters (e.g. the recession time scales of the different reservoirs, which represent subsurface flow resistances through the system). We will include this in the discussion of the revised manuscript.*

**Comment:**

L 208 „we expect …" First off, I appreciate the formulation of hypotheses. I wold only statethem at the end of the introduction however. Also, where hypotheses are formulated, it can be highly confusing to leave ambiguity between transpiration, bare soil evaporation, or evapotranspiration for the two combined. Please specify. Finally, where does the hypothesis come from? Please add references.

*Reply*

*We will clarify in the revised version of the manuscript that we use the term evaporation to represent all different evaporative fluxes (interception, soil evaporation, transpiration). We follow the terminology proposed by Savenije (2004) and Miralles et al. (2020), where evaporation instead of evapotranspiration is used to refer to all evaporative fluxes.*

*In the revised version of the manuscript, we will also include the references of Fenicia et al. (2009), Teuling et al. (2019) and Stephens et al. (2021).*

**Comment:**

L 222 „match expectations .. ": Unclear formulation, please be more specific on what typeof expectation.

*Reply*

*Also here, we will rephrase that it is about the expectation that catchments scatter closely around the analytical Budyko curve, suggesting a co-evolution of vegetation and soils with climate.*

**Comment:**

L 279 What is meant with „imposing"? I do not understand what is done here.

*Reply*

*Agree, we will rephrase to explain that we added the difference between the simulated 2K and historical water deficits to the observed historical climate deficits (E-OBS data). This was done to account for the bias between the simulated and observed historical climate data. Perhaps this technical detail can be moved to the supplement of the revised version of the manuscript not to confuse the reader.*

**Comment:**

Table 1

I am confused about the last Last column: Over what sample is the max and min taken?Why do those max and min not appear on the first two lines?

*Reply*

*See also our reply to the previous comment, for the 2K scenario, we add the difference between the simulated 2K and historical water deficits to the observed historical climate deficits to account for the bias between the simulated and observed historical climate data. In the revised version of the manuscript, we will move these technical details on this "bias-correction" to the supplement in order not to confuse the reader in the main storyline.*

**Comment:**

L 298: See my previous notes on abbreviations. Better would be „Hydrological model:xxxx"

*Reply*

*Agreed, we will change this.*

**Comment:**

L 303-304: The values appear arbitrary, and maybe are explained in the references. Isuggest adding an explanation of their origin, so that the reader can understand the general idea without need to refer elsewhere.

*Reply*

*Agreed, these values were retrieved from the study of Gharari et al. (2011), we will clarify this in the revised version of the manuscript.*

**Comment:**

L 306: Am I understanding correctly that hillslope vs plateau was derived from a vegetation map? If yes, please spell this out more clearly, it is very opaque from the current description. Also, in other words, the main difference between hillslope and plateau, which is the consideration of deep drainage, depends on the vegetation as well, with agricultural areas allowing for deep drainage and forested areas (per definition of thelocations) does not? How does this affect the model results? This needs to enter the discussion.

*Reply*

*In the revised version of the manuscript, we will make sure to clarify that the Hydrological Response*

*Units hillslope, plateau and wetland are first derived from topographical information based on thresholds for the Height Above the Nearest Drain and slope. As additional step, we associate forest with hillslope and agricultural area with plateau using the land use map. Groundwater recharge occurs both in the plateau and the hillslope HRU through preferential recharge from the root-zone storage. In the plateau class, there is also recharge through percolation from the root-zone to the groundwater. In the land-use scenarios, we did not change the percentages HRU as we are using a top-down approach to estimate the changes in runoff coefficient through trajectories in the Budyko space, which does not include detailed information on the exact spatial extent of change. We do not expect a large effect of this limitation on our results, but we will mention it in the discussion.*

**Comment:**

L 307 - 311: Important information is missing. Important information would be, what happens, when the root zone storage runs dry? Does this affect ET at all? What happens when ET cannot be satisfied?

Given the general topic of the paper, it needs to be clearly explained how vegetation affects hydrology in this model, especially E_A. At this point I am assuming that E_A is imposed either from observations or regional climate model and further modified to accommodate the different runoff coefficients that are taken to represent the vegetation cover? Later (in the discussion) I am learning that water availability actually affects E_A and I am back at point zero. This section really needs attention.

**Reply**

*In the revised version of the manuscript, we will clearly explain that we use a standard formulation to express water stress. Evaporation from the root-zone ($E_R$) is reduced when the storage is below a certain threshold. The detailed equations are provided in the Supplement (Table S4).*
*The model is forced with potential evaporation and actual evaporation is an output of the model.*

**Comment:**

L 324-326: We learn elsewhere that this corresponds to plateau corresponds to agriculture and hill slope to forest. Please repeat this here. This is an important part of the study.

**Reply**

*This is a good point, the HRU are actually derived both from topographical and land use data, we will repeat it here.*

**Comment:**

L 339-340: In other words, potential interactions between the model parameters are neglected? Was this tested?

*Reply*

*We agree that it is not unlikely that future changes may also influence other system characteristics. However, the mutual interactions between parameters are so far unknown and were therefore not explicitly considered. We did use an ensemble of parameter sets to somehow account for the uncertainty in model parameters and the possibility that parameters compensate for each other due to simplistic process representation. In many studies, the hydrological model used for future simulations remains completely identical to the model structure derived for historical conditions. In our study, we perform a controlled experiment to test the sensitivity of changing the root-zone storage capacity, which we can estimate from the future climate data. We will discuss this in the limitation section of the discussion of the revised manuscript.*

**Comment:**

L 358: Here and in D: Does the forcing for ET change as a result of the land use change?

*Reply*

*The potential evaporation used as forcing was calculated with the Makkink formula and we did not change it as a result of the land-use change. We will mention this in the limitation section of the discussion.*

**Comment:**

L 378-379: Sounds like interpretation and this should go to discussion.

*Reply*

*We agree that this sentence could be perceived as discussion. However, it is always difficult to clearly separate results from discussion. Here, we think it provides guidance to the reader to place the results in a broader context*

**Comment:**

L 389-399: Maybe merge the sections on root zone storage across scenarios A-D?

*Reply*

*We agree that splitting the results with separate sections for each scenario leads to some repetitions. However, we also think it increases the clarity to treat each scenario separately. In the revised version of the manuscript, we will critically reflect on how we can further clarify the structure to present the different scenarios.*

**Comment:**

L 395-396: Would be good to have an overview table with the climate conditions (aridity, seasonality, P, E_pot) for all scenarios, including separate listing of E_pot and E_A for the scenarios.

*Reply*

*We thank the reviewer for this suggestion, and we will include such a table in the revised version of the manuscript.*

**Comment:**

L 445-455: Was very confused about how the E_A was obtained. It is a model output or forcing?

*Reply*

*In the revised version of the manuscript, we will clarify that actual evaporation is model output. The forcing variables of the model are potential evaporation, precipitation and temperature.*

**Comment:**

L 467-468: „result of soil moisture stress in the root-zone" This is not mentioned in the model description and is absolutely a must. Also, please report on times of soil water stress in the model scenarios.

*Reply*

*As also mentioned in earlier replies, we will explain that we use a standard formulation to represent water stress, as described in the model equations of Table S4.*

**Comment:**

Discussion: I find it more logical that the limitations are stated first, followed by interoperation and implications last.

*Reply*

*This is perhaps a matter of taste, but we think limitations can also be read as an outlook for future work and we therefore think it might better fit after the implication section.*

**Comment:**

L 501-503: This sentence can be erased without loosing information.

*Reply*

*Agree, we will remove this sentence.*

**Comment:**

L 547-548: Also rooting depth is species specific, and mono-cultures would have limited capacity to adapt.

*Reply*

*We agree. As also mentioned in one of the earlier replies, there is a distinction between rooting depth and root-zone storage capacity (i.e. the water volume accessible to the roots of vegetation for transpiration).*

**Comment:**

Figure 2: The arrow with script \Delta \omega is misleading. What is shown is \Delta (E_A / P),which is really not the same.

*Reply*

*Thank you for this good point. We will remove the arrow from the schematization and revise the caption.*

**Comment:**

Figure 3: See above major comments. The important difference is in how the interaction with groundwater is accounted for in the different slope positions, which are at the same timedirectly linked to vegetation cover. This is an important detail and should be made obvious. In contrast, the remainder of the Figure is not very important and could in my opinion go to the appendix.

*Reply*

*Thank you for this suggestion. We will emphasize the differences in the caption and we agree to move the Figure to the Supplement. This way, it will be clearly connected to the model equations.*

References

Bouaziz, L. J. E., Fenicia, F., Thirel, G., De Boer-Euser, T., Buitink, J., Brauer, C. C., et al. (2021). Behind the scenes of streamflow model performance. *Hydrology and Earth System Sciences*, *25*(2), 1069–1095. https://doi.org/10.5194/hess-25-1069-2021

Fenicia, F., Savenije, H. H. G., & Avdeeva, Y. (2009). Anomaly in the rainfall-runoff behaviour of the Meuse catchment. Climate, land-use, or land-use management? *Hydrology and Earth System Sciences*, *13*(9), 1727–1737. https://doi.org/10.5194/hess-13-1727-2009

Gentine, P., D'Odorico, P., Lintner, B. R., Sivandran, G., & Salvucci, G. (2012). Interdependence of climate, soil, and vegetation as constrained by the Budyko curve. *Geophysical Research Letters*, *39*(19), 2–7. https://doi.org/10.1029/2012GL053492

Gharari, S., Hrachowitz, M., Fenicia, F., & Savenije, H. H. G. (2011). Hydrological landscape classification: Investigating the performance of HAND based landscape classifications in a central European meso-scale catchment. *Hydrology and Earth System Sciences*, *15*(11), 3275–3291. https://doi.org/10.5194/hess-15-3275-2011

Guswa, A. J. (2008). The influence of climate on root depth: A carbon cost-benefit analysis. *Water Resources Research*, *44*(2), 1–11. https://doi.org/10.1029/2007WR006384

Harman, C., & Troch, P. A. (2014). What makes Darwinian hydrology "darwinian"? Asking a different kind of question about landscapes. *Hydrology and Earth System Sciences*, *18*(2), 417–433. https://doi.org/10.5194/hess-18-417-2014

Hrachowitz, M., Stockinger, M., Coenders-Gerrits, M., van der Ent, R., Bogena, H., Lücke, A., & Stumpp, C. (2020). Deforestation reduces the vegetation-accessible water storage in the unsaturated soil and affects catchment travel time distributions and young water fractions. *Hydrology and Earth System Sciences*, *i*(June), 1–43. https://doi.org/10.5194/hess-2020-293

Jasechko, S. (2018). Plants turn on the tap. Nature Climate Change, 8, 560–563.

Kervyn, T., Jacquemin, F., Branquart, E., Delahaye, L., Dufrêne, M., & Claessens, H. (2014). Les forêts anciennes en Wallonie. 2ème partie : Cartographie. *Forêt Wallonne*, *133*, 38–52.

Miralles, D. G., Brutsaert, W., Dolman, A. J., & Gash, J. H. (2020). On the use of the term "Evapotranspiration." *Earth and Space Science Open Archive*, 8. https://doi.org/10.1002/essoar.10503229.1

Savenije, H. H. G. (2004). The importance of interception and why we should delete the term evapotranspiration from our vocabulary. *Hydrological Processes*, *18*(8), 1507–1511. https://doi.org/10.1002/hyp.5563

Schymanski, S. J., Sivapalan, M., Roderick, M. L., Beringer, J., & Hutley, L. B. (2008). An optimality-based model of the coupled soil moisture and root dynamics. *Hydrology and Earth System Sciences*, *12*(3), 913–932. https://doi.org/10.5194/hess-12-913-2008

Stephens, C. M., Lall, U., Johnson, F. M., & Marshall, L. A. (2021). Landscape changes and their hydrologic effects: Interactions and feedbacks across scales. *Earth-Science Reviews*, *212*(September 2020), 103466. https://doi.org/10.1016/j.earscirev.2020.103466

Teuling, A. J., De Badts, E. A. G., Jansen, F. A., Fuchs, R., Buitink, J., Van Dijke, A. J. H., & Sterling, S. M. (2019). Climate change, reforestation/afforestation, and urbanization impacts on evapotranspiration and streamflow in Europe. *Hydrology and Earth System Sciences*, *23*(9), 3631–3652. https://doi.org/10.5194/hess-23-3631-2019

Yang, Y., Donohue, R. J., & McVicar, T. R. (2016). Global estimation of effective plant rooting depth: Implications for hydrological modeling. *Water Resources Research*, *52*(10), 8260–8276. https://doi.org/10.1002/2016WR019392

---

## Author Comment (AC3)

*Reply to Anonymous Referee # 3*

**Overall assessment by Referee #3:**

This is a very interesting study on the possible implications of ecosystem root-zone storage capacity changes induced by vegetation adaptation to climate change. The authors use a top-down approach based on the Budyko model. I believe that the study is novel and the insight provided by the study is valuable. The methods are innovative and useful for the Hydrology and earth system science community. However, there are several aspects in the methodology that need to be further explained/clarified to improve the quality of this contribution.

*Reply:*

*We thank Referee #3 for his/her positive assessment of our manuscript. We provide a reply to each of the valuable comments below.*

**Comment 1:**

Lines 144-145 refers to a monthly bias-correction factor applied to improve the consistency between the "E-OBS dataset in the center of the basin when compared to an operational dataset" which is "based on local precipitation data provided by the Service Public de Wallonie for the period 2005-2017". Though there are some additional details in the supplement this comment is very vague here, so it would be good to add some further clarification on the rationale for the use of the bias- correction factor, and why it "improves consistency".

*Reply 1:*

*We agree with this suggestion and will clarify in the main text that we correct the E-OBS dataset to better represent the local precipitation data provided by the Service Public de Wallonie. As also detailed in the Supplement, we use a monthly correction factor in the center part of the basin because the E-OBS data underestimates the interpolated station data with more than 20%.*

**Comment 2:**

Lines 227-228 state: "The water-balance method requires daily time series of precipitation, potential evaporation and a long-term runoff coefficient to estimate transpiration, as it depletes the root-zone storage during dry spells." Dry spells can be interpreted as interannual periods (a dry spell could potentially last more than one year in certain regions), but here you are only considering seasonal dry periods... so please clarify.

*Reply 2:*

*The reviewer is completely correct. We will clarify that, in our study area, the dry spells are seasonal as storage deficits become zero again in the fall and winter when excess precipitation drains away as direct runoff or recharge.*

**Comment 3:**

Lines 231-235: The explanation on the use of equation (4) and the estimation of the associated variables is not clear. The problem might be that at this stage in the manuscript, the model used for

the estimation of the hydrologic variables has not been presented yet (it is later presented in section 4.2 and schematized in Figure 3). It is then difficult for the reader to understand how is PE estimated based on the other variables in this equation (as EI and SI are not available from observations). It is therefore important to explain how EI and SI are estimated (here and not later, perhaps linking to the use of the model here, mentioning that the details will be described later). Please also explain if there is an implied iterative process. That is, in order to estimate EI and SI from the model (shown also in Figure 3), the value of Sr, max needs to be set, right? But it is obtained after using equation 4 (which uses the results of the model). I find the explanation of the methodology in this aspect unclear, so this needs to be further clarified.

*Reply 3:*

*The estimation of the root-zone storage capacity with the water-balance approach is an independent step, which is not necessarily linked to the use of a specific model structure. However, it is correct that we use the same interception module as in the model to estimate the interception evaporation in the water-balance approach. The module consists of a reservoir with a maximum interception storage $I_{max}$ to determine effective precipitation ($P_E = max(0, S_I - I_{max})/dt)$) and interception evaporation ($E_I = min (E_P, (S_I - I_{max})/dt)$). We will include these formulas in the revised version of the manuscript. The value of $S_{R,max}$ does not need to be set to run the interception module to estimate $E_I$ and $S_I$, as interception processes occur before precipitation reaches the root-zone. Therefore, after estimating the effective precipitation, the value of $S_{R,max}$ in the water-balance approach is estimated, which does not require an iterative process. We will clarify this part in the revised version of the manuscript.*

**Comment 4:**

Line 249: I think it should be "By fitting the extreme value distribution of Gumbel to the series of annual maximum storage deficits"

*Reply 4:*

*Yes, thank you, we will change this!*

**Comment 5:**

Lines 249: Why Gumbel?

*Reply 5:*

*We used the Gumbel distribution as it is frequently used for estimating hydrological extremes. In particular, it was previously shown to be a suitable choice for the estimation of the root-zone storage capacity through the water-balance approach by several other studies (Gao et al., 2014; Nijzink et al., 2016; de Boer-Euser et al.; 2016, Wang-Erlandsson et al., 2016; Bouaziz et al., 2020; Hrachowitz et al., 2020). We will clarify this in the revised version of the manuscript.*

**Comment 6:**

Line 271. What do you mean by "native" simulated … ?

*Reply 6:*

*With "native", we mean that we did not apply a bias-correction to the simulated historical climate data. We will clarify this in the revised version of the manuscript.*

**Comment 7:**

Figure 5 a is not clear (difficult to visualize). Perhaps a change on the colour scheme used for the lines (more contrasting colours) could help.

*Reply 7:*

*We agree with the reviewer that all the curves in Fig 5a are difficult to visualize. The color scheme used in Figure 5 is consistent with the color scheme used in the other Figures. However, we think it might be sufficient to only show the dashed curves representing the median $\omega_{obs}$-values and remove the 35 curves of the other catchments that are indeed not clearly visible in the Figure. We will adapt this in the revised version of the manuscript.*

**Comment 8:**

Line 421 states: "The ensemble of parameter sets retained as feasible after calibration mimics the observed hydrograph…".  I think that you are trying to say: The simulated values of Q obtained using "the ensemble of parameter sets retained as feasible after calibration mimics the observed hydrograph…".

*Reply 8:*

*Yes, this is correct, thank you, we will adapt this.*

References

de Boer-Euser, T., McMillan, H. K., Hrachowitz, M., Winsemius, H. C., & Savenije, H. H. G. (2016). Influence of soil and climate on root zone storage capacity. *Water Resources Research*. https://doi.org/10.1002/2015WR018115

Bouaziz, L. J. E., Steele-Dunne, S. C., Schellekens, J., Weerts, A. H., Stam, J., Sprokkereef, E., et al. (2020). Improved understanding of the link between catchment-scale vegetation accessible storage and satellite-derived Soil Water Index. *Water Resources Research*. https://doi.org/10.1029/2019WR026365

Gao, H., Hrachowitz, M., Schymanski, S. J., Fenicia, F., Sriwongsitanon, N., & Savenije, H. H. G. (2014). Climate controls how ecosystems size the root zone storage capacity at catchment scale. *Geophysical Research Letters*, *41*(22), 7916–7923. https://doi.org/10.1002/2014GL061668

Hrachowitz, M., Stockinger, M., Coenders-Gerrits, M., van der Ent, R., Bogena, H., Lücke, A., & Stumpp, C. (2020). Deforestation reduces the vegetation-accessible water storage in the unsaturated  soil and affects catchment travel time distributions and young water fractions. *Hydrology and Earth System Sciences*, *i*(June), 1–43. https://doi.org/10.5194/hess-2020-293

Nijzink, R., Hutton, C., Pechlivanidis, I., Capell, R., Arheimer, B., Freer, J., et al. (2016). The evolution of root-zone moisture capacities after deforestation: A step towards hydrological predictions under change? *Hydrology and Earth System Sciences*, *20*(12), 4775–4799. https://doi.org/10.5194/hess-20-4775-2016

Wang-Erlandsson, L., Bastiaanssen, W. G. M., Gao, H., Jägermeyr, J., Senay, G. B., Van Dijk, A. I. J. M., et al. (2016). Global root zone storage capacity from satellite-based evaporation. *Hydrology and Earth System Sciences*, *20*(4), 1459–1481. https://doi.org/10.5194/hess-20-1459-2016

---

## Referee Report (RR1)

**Review of the manuscript "Ecosystem adaptation to climate change: the sensitivity of hydrological predictions to time-dynamic model parameters", ID hess-2021-204**

I want to thank the authors for their detailed reply to my comments. I think that they have been well addressed in the revised version of the manuscript which further improved the manuscript.

I only have some minor suggestions/comments left:

- L. 19-21: Revise this sentence for better readability ("with in the non-stationary scenarios up to…" is hard to follow)
- L. 740-743: "This root-accessible water volume is independent of soil type …", is there a reference for these two sentences?
- L. 793: "increasing its root-zone storage capacity **by** 34%" -> replace "with" with "by"

**Figure 2:** This graph is a nice addition to the method section, as it nicely supports understanding of the methods workflow. However, I think that it should be revised a bit for more structure, readability and consistency. The workflow/concept presented in the figure should also be clear to someone that has not read the full method section. I have a few suggestions that might help revising the figure:

- Text:
    - Could be a shorter and more consistent across the columns to make clear how climate and input parameters differ between columns.
        - E.g. "fixed $S_{R,max,A}$ parameter" (col 3) vs. "$S_{R,max,A}$" (col 4) is inconsistent
        - Maybe instead of text, you could put two bullet points *Climate: xxx* and *Root zone storage capacity: xxx* in each modeling column. This would make it easy to compare at one glance in which ways columns differ from each other?
    - Headings in the sub columns might help for structure of workflow
        - E.g. col (3) Calibration/Model calibration, col (4) Evaluation/Validation
- 4 Scenarios:
    - It is a bit confusing to have the historical scenario from (4) also in each of the scenario description of (5). At a first glance, it looks like the reader has to understand 8 instead of 4 scenarios until they figure out that 4 of them are the same. I think the concept of 4 climate and land-use change scenarios that are all compared to the same historical scenario could be made clearer. You could, for example, have the scenario description of the historical scenario in col (4) and the four 2K scenarios in column (5). Then, in column (6) you explain that you compared all scenarios from (5) to the scenario from (4)
- Currently, there is only one arrow linking the output $S_{RmaxA}$ from col (2) with column (3). But shouldn't there also be arrows linking the 4 Q/P Outputs from col (1) with col (2) and the remaining 3 $S_{Rmax}$ B-D from col (2) with the scenarios in col (5)? I realize that in the current structure of the plot this might be difficult. Maybe it's easier if you flipped the plot structure to a top-bottom instead of left-right structure?
- Shading and boxes could be improved to support the workflow better:
    - Too many shades of gray (e.g. headings are different from columns) make it difficult to see the clear main workflow from "Estimate Srmax" -> Modeling -> Change evaluation.

---

## Author Response (AR2)

*Dear Editor, dear Erwin Zehe,*

*We thank you for the promising words and are very glad to submit the revised manuscript after incorporating the minor revisions made by the referees, which further contribute to improving the manuscript.*

*In the revision, we mainly adapted Figure 2 as suggested by Referee #1 and we have revised the discussion based on the comments made by Referee #2.*

*We look forward to hearing from you.*

*Best wishes,*

*Laurène Bouaziz and co-authors.*

**Overall assessment by Referee #1:**

I want to thank the authors for their detailed reply to my comments. I think that they have been well addressed in the revised version of the manuscript which further improved the manuscript.

*Reply:*

*We thank Referee #1 for the positive words on our revised manuscript. We have addressed the remaining minor suggestions and reply to them below.*

**Comment 1:**

L. 19-21: Revise this sentence for better readability ("with in the non-stationary scenarios upto…" is hard to follow)

*Reply 1:*

*Agreed, we have split this sentence in two to increase readability.*

**Comment 2:**

L. 740-743: "This root-accessible water volume is independent of soil type …", is there a reference for these two sentences?

*Reply 2:*

*We have added relevant references for these sentences.*

**Comment 3:**

- L. 793: "increasing its root-zone storage capacity **by** 34%" -> replace "with" with "by"

*Reply 3:*

*Agreed, we have corrected with "by".*

**Comment 4:**

**Figure 2:** This graph is a nice addition to the method section, as it nicely supports understanding of the methods workflow. However, I think that it should be revised a bit for more structure, readability and consistency. The workflow/concept presented in the figure should also be clear to someone that has not read the full method section. I have a few suggestions that might help revisingthe figure:

- Text:

- o Could be a shorter and more consistent across the columns to make clear howclimate and input parameters differ between columns.
  - ▪ E.g. "fixed $S_{R,max,A}$ parameter" (col 3) vs. "$S_{R,max,A}$" (col 4) is inconsistent
  - ▪ Maybe instead of text, you could put two bullet points *Climate: xxx* and *Rootzone storage capacity: xxx* in each modeling column. This would make it easyto compare at one glance in which ways columns differ from each other?
  - o Headings in the sub columns might help for structure of workflow
    - ▪ E.g. col (3) Calibration/Model calibration, col (4) Evaluation/Validation
- 4 Scenarios:
  - o It is a bit confusing to have the historical scenario from (4) also in each of the scenario description of (5). At a first glance, it looks like the reader has to understand 8 instead of 4 scenarios until they figure out that 4 of them are the same. I think the concept of 4 climate and land-use change scenarios that are all compared to the same historical scenario could be made clearer. You could, for example, have the scenario description of the historical scenario in col (4) and the four 2K scenarios in column (5). Then, in column (6) you explain that you comparedall scenarios from (5) to the scenario from (4)
- Currently, there is only one arrow linking the output $S_{RmaxA}$ from col (2) with column (3). But shouldn't there also be arrows linking the 4 Q/P Outputs from col (1) with col (2) and the remaining 3 $S_{Rmax}$ B-D from col (2) with the scenarios in col (5)? I realize that in the current structure of the plot this might be difficult. Maybe it's easier if you flipped the plot structureto a top-bottom instead of left-right structure?
- Shading and boxes could be improved to support the workflow better:
  - o Too many shades of gray (e.g. headings are different from columns) make it difficultto see the clear main workflow from "Estimate Srmax" -> Modeling -> Change evaluation.

*Reply 4:*

*We thank the referee for the good suggestions to improve the consistency and readability of the Figure, we have been playing around with the design and have made the following changes:*

- *We think it is indeed a very good idea to flip the layout, this also reduces the shades of grey and makes it easier to add titles for each step of the procedure*
- *We rearranged the 4 scenarios and only show the historical schema once.*
- *We removed the arrow, as it should already be clear from the text that we are using results from a previous step.*

*Reply to Anonymous Referee # 2*

**Overall assessment by Referee #2:**

I have already reviewed the previous version of this manuscript. In the revision the authors have really improved the accessibility of the manuscript and included much more interpretation of the results in an extended discussion.

The Methods section is much improved, the collection of all the model information in Appendix 3 works well. Thanks for including some details, e.g. a short note on how transpiration decreases when root zone storage dries. Those pointers will really help readers (like me) to focus on the main message and not be distracted by searching. The discussion has much increased in size and enhances the quality of the manuscript. Most of my comments have been satisfactorily addressed or ruled out. Mainly on the new discussion, I have some more comments below. I propose to include a couple more aspects. Also I propose to bring some of the sections, especially in the „limitations"-part, to a resolution, e.g. how can those be tackled in the future?

I believe only minor revisions are required.

*Reply:*

*We thank Referee #2 for positively reassessing our revised manuscript. We have addressed the remaining constructive comments and reply to them below.*

**Comment 1:**

L 308 „We repeat this random sampling several times .. several model parameters"
Ok, I did not understand this in the previous version. It remains unclear at this point: How much is several, and what happens to the sample? Do you proceed with each individual realization or some aggregated (average) runoff coefficient?

*Reply 1:*

*We have clarified that we repeated the random sampling seven times and that we indeed work with the ensemble of each individual realization in the subsequent analyses, e.g. when plotting the uncertainty bands in Fig 4c, 7 and 8.*

**Comment 2:**

L 279-281: It is really mostly the rooting depth that will be adaptable. I propose to include a short comment here, to help the reader make the association between rooting storage and rooting depth, which now change with vegetation type.

*Reply 2:*

*This is a good suggestion, we have included a short sentence in the following paragraph (line 288 in tracked change manuscript) to clarify that the adaptation of the root-zone storage capacity mainly relates to changes in the rooting system (rooting depth and/or rooting density).*

**Comment 3:**

Discussion: The role of the investigation of land use change is still a bit opaque in the manuscript. The Methods section says on lines 224-226: „These scenarios are meant as a sensitivity analysis in the spirit of trading space-for-time (Singh et al., 2011) to evaluate the effect of potential future land-use management on the overall water balance."

I have interpret this to say that climate change scenarios will be evaluated in the light of the additional affect of land use change? This sounds very reasonable to me. But I wondered, why this idea was not taken up in the discussion? The „implications" section of the discussion revolves mostly around climate change and I missed the reflection on the effects of the vegetation transformation. Can you more specifically return to the comment made earlier in the manuscript?

*Reply 3:*

*Indeed, we evaluate the effect of the additional effect of potential land use change and the referee is completely right that it would be good to come back to it in the Implication section of the discussion. Now, the relatively limited additional effects of land-use change on the hydrological response are mentioned in section 6.2.2 "On the potential land-use change scenarios", but this is indeed part of the Limitation section. We have added a sentence (lines 537-539 of the tracked change manuscript) on this also at the end of the first paragraph of the Implication section of the discussion.*

**Comment 4:**

L 553-562 & L 565-579: Here you discuss one the one hand that present management practices affect the future and second the limitation that we do not know whether plants will be readily adapt to the root zone storages required in the altered climate. It would have been nice to have this connected a bit more, and given some idea on why the investigation is nevertheless useful. For example, it proposes an envelope of the hydrological response, which is based on the best of our knowledge. Furthermore, do you agree that the method can also be used to estimate the degree of adaptation in existing catchments? This would be one research outlook to improving the predictions.

*Reply 4:*

*Thank you, we agree with this very good suggestion. We have added a sentence (lines 569-570 of the tracked change manuscript) to make it more explicit that indeed, despite the uncertainty, the study is useful as it quantifies an envelope of the hydrological response when we consider ecosystem adaptation in response to climate change and that the method can be used to estimate the potential adaptation of ecosystems in catchments.*

**Comment 5:**

L 609-612: I believe this has been added in response to one of my comments. It now comes a bit out of context here, and I propose to erase it. Mentioning the constant potential evaporation in the methods section works well. Adding a comment in the Methods section that swapping omega does not affect the HRUs would be helpful for readers like me.

*Reply 5:*

*Agreed, we have moved these sentences of the discussion to method section 4.4 (Hydrological change*

*evaluation).*

**Comment 6:**

L 639-640: „very interesting" is a bit vague. Can you be a bit more specific? Why would it be and do you have an intuition how the water fluxes may change with adaptive roots compared to the original HTESSEL runs? Maybe also have a look at this publication in this context:

Stevens, D., Miranda, P. M. A., Orth, R., Boussetta, S., Balsamo, G., & Dutra, E. (2020). Sensitivity of surface fluxes in the ECMWF land surface model to the remotely sensed leaf area index and root distribution: Evaluation with tower flux data. Atmosphere, 11(12), 1–19. https://doi.org/10.3390/atmos11121362

*Reply 6:*

*Thank you for this suggestion and the provided reference. We have clarified that the implementation of a time-dynamic root-zone storage capacity in land surface model could improve the estimation of evaporative fluxes (and by extension also streamflow). However, quantifying how the water fluxes may change with adaptative root-zone storage capacity in the HTESSEL runs would really require a dedicated and thorough analysis for the study area, which is outside the scope of the current work.*

**Comment 7:**

L 650-651: „may allow us to estimate future ω or root-zone storage capacity values in a region where the future climate may resemble today's climate elsewhere"
I am admittedly not sure where this is going. Can you be more specific on the expected insights? Do you propose to test how omega relates to root storage in different climates? This would be very nice and eventually help to validate one of the major assumptions of the manuscript, which is that omega changes due to vegetation and not climate.

Finally, I would really like to read a short discussion on how changes in climate dynamics would affect omega independently of vegetation in a future climate?

*Reply 7:*

*What we mean here is to develop data-based models/relations which relate either omega and/or the root-zone storage capacity in a large sample of catchments to relevant climatic, topographic, soil or land-use indicators. This would enable us to predict how omega or the root-zone storage capacity may change in response to changes in the combination of these various indicators. This could be a more robust, data-based method to estimate potential changes in root-zone storage capacity or omega values. We have clarified this in the revised manuscript.*

*In our study, we indeed assume that changes in omega are mainly related to changes in vegetation in a future climate. While we already mention in Section 6.2.2 "On the potential land use change scenarios" that omega values are likely a manifestation of multiple climatic, landscape and vegetation characteristics, we have now more strongly emphasized that climate variability, CO2-vegetation feedbacks and/or anthropogenic modifications may also influence omega values (Berghuijs et al., 2017).*

**Comment 8:**

Appendix S3: It is good to have all the model related details collected in this section. Very clear now.

It seems that some of the equations in Table S5 are potentially not correct? Should it not read $dS_w/dt$ instead of $S_w/dt$? This is repeated in several lines. Also in the line relating to $E_R$: I am not sure I saw a definition of $S_u$.

*Reply 8:*

*Great to hear that it is now very clear to have all the information related to the model in the Appendix!*

*Thank you for pointing out the $S_U$ which should indeed be replaced by $S_R$. However, the presence of $S_w/dt$ is not incorrect, as explicitly mentioning the time step of the model in the equations ensures that the units of the stores S [mm] divided by the timestep [d] match the units of the fluxes [mm/d].*

*References:*

*Berghuijs, W.R., Larsen, J.R., Van Emmerik, T.H. and Woods, R.A., 2017. A global assessment of runoff sensitivity to changes in precipitation, potential evaporation, and other factors. Water Resources Research, 53(10), pp.8475-8486.*

*Reply to Patricia Saco (Referee #3)*

**Overall assessment by Patricia Saco:**

I believe that the authors have carefully and thoroughly addressed all the reviewers comments. The revised manuscript has been greatly improved in the revision process.

*Reply:*

*We would like to thank Patricia Saco for her very positive assessment of our revised manuscript.*

[revised manuscript text omitted]

---

## Author Response (AR3)

*Dear Editor, dear Erwin Zehe,*

*We are delighted to hear this excellent news and we also thank you for handling our manuscript.*

*Best wishes,*

*Laurène Bouaziz and co-authors.*